**EMBO**
*reports*

# A conserved ion channel function of STING mediates noncanonical autophagy and cell death

Jinrui Xun [1,2,7], Zhichao Zhang [3,7], Bo Lv [2,7], Defen Lu [3,7], Haoxiang Yang [2], Guijun Shang [3,4,5 ✉] & Jay Xiaojun Tan [2,6 ✉]

## Abstract

The cGAS/STING pathway triggers inflammation upon diverse cellular stresses such as infection, cellular damage, aging, and diseases. STING also triggers noncanonical autophagy, involving LC3 lipidation on STING vesicles through the V-ATPase-ATG16L1 axis, as well as induces cell death. Although the proton pump V-ATPase senses organelle deacidification in other contexts, it is unclear how STING activates V-ATPase for noncanonical autophagy. Here we report a conserved channel function of STING in proton efflux and vesicle deacidification. STING activation induces an electron-sparse pore in its transmembrane domain, which mediates proton flux in vitro and the deacidification of post-Golgi STING vesicles in cells. A chemical ligand of STING, C53, which binds to and blocks its channel, strongly inhibits STING-mediated proton flux in vitro. C53 fully blocks STING trafficking from the ER to the Golgi, but adding C53 after STING arrives at the Golgi allows for selective inhibition of STING-dependent vesicle deacidification, LC3 lipidation, and cell death, without affecting trafficking. The discovery of STING as a channel opens new opportunities for selective targeting of canonical and noncanonical STING functions.

**Keywords** STING; Ion Channel; Noncanonical Autophagy; Vesicle Deacidification; Membrane Trafficking
**Subject Categories** Autophagy & Cell Death; Membranes & Trafficking

## Introduction

The cyclic GMP-AMP (cGAMP) synthase (cGAS)/stimulator of interferon genes (STING) innate immunity pathway is critical in host defense against microbial infection (Ishikawa and Barber, 2008; Ishikawa et al, 2009; Sun et al, 2009; Zhong et al, 2008). Besides infection-triggered inflammation, the cGAS/STING path-

way also promotes sterile inflammation in many other conditions involving cytosolic exposure of self-DNA, such as cell damage, senescence, diseases, and normal aging (Decout et al, 2021; Motwani et al, 2019). Upon recognition of abnormal cytosolic DNA, cGAS produces a second messenger cGAMP, which binds to and activates STING on the endoplasmic reticulum (ER) (Sun et al, 2013; Wu et al, 2013). The cGAMP-bound STING undergoes robust trafficking from the ER to the Golgi region and finally forms perinuclear vesicle clusters where TBK1 is recruited to activate both IRF3 and NF-κB, transcription factors that upregulate the expression of type I interferons (IFN-I) and inflammatory cytokines (Tan et al, 2018).

STING is also known to restrict microbial infection through noncanonical autophagy, which is independent of autophagy proteins upstream of the ATG5/ATG12/ATG16 complex (Gui et al, 2019; Liu et al, 2019; Moretti et al, 2017). An extensively studied form of noncanonical autophagy is known as the conjugation of LC3/ATG8 to single membranes or CASM (Durgan and Florey, 2022). CASM induction is mediated through the direct recruitment of ATG16 by the vacuolar-type ATPase (V-ATPase), a proton pump that senses organelle deacidification upon challenges from weak base chemicals, proton ionophores, or lysosomal membrane damage (Cross et al, 2023; Tan and Finkel, 2022; Wang et al, 2022; Xu et al, 2019). A recent study from the Youle group identified V-ATPase as a downstream effector of STING for noncanonical LC3 lipidation on single membrane STING vesicles (Fischer et al, 2020). However, it was still unclear how STING activates V-ATPase.

Here, we report that STING-induced noncanonical autophagy depends on an unexpected ion channel function of STING. Ligand-bound STING forms an electron-sparse pore in its transmembrane domain that causes proton efflux from post-Golgi STING vesicles and subsequent vesicle deacidification. This is required for STING-dependent LC3 lipidation and cell death, both of which can be abolished by C53, a small molecule compound that blocks the transmembrane channel of STING. This new molecular function of STING resembles other CASM-inducing stresses that deacidify endolysosomes or phagosomes.

[1]Xiangya School of Medicine, Central South University, Changsha, China. [2]Aging Institute, University of Pittsburgh School of Medicine/University of Pittsburgh Medical Center, Pittsburgh, PA, USA. [3]College of Life Sciences, Shanxi Agricultural University, Taiyuan, China. [4]The Key Laboratory of Medical Molecular Cell Biology of Shanxi Province, Institutes of Biomedical Sciences, Shanxi University, Taiyuan, China. [5]Shanxi Provincial Key Laboratory of Protein Structure Determination, SAARI, Taiyuan, China. [6]Department of Cell Biology, University of Pittsburgh School of Medicine, Pittsburgh, PA, USA. [7]These authors contributed equally: Jinrui Xun, Zhichao Zhang, Bo Lv, Defen Lu. ✉E-mail: gjshang@gmail.com; Jay.Tan@pitt.edu

# Results

## Activated STING forms a potential channel in its transmembrane region

Our recent structural study of apo-STING and cGAMP/STING complex provided new mechanistic insights into STING autoinhibition and activation (Hussain et al, 2022; Liu et al, 2023b). Notably, upon cGAMP binding, the ligand binding domain (LBD) of STING contracted, whereas the transmembrane domain (TMD) underwent dilation. After a thorough analysis of the conformational changes in the TMD from the apo and cGAMP-bound structures of human STING, a ligand-induced transmembrane pore with a radius of ~1.71 Å at its narrowest part was revealed (Fig. 1A,B). In contrast, the pore radius was ~0.86 Å at its bottleneck in the apo-structure (Fig. 1A). The ligand-induced pore space was electron-sparse (Liu et al, 2023b), suggesting that it is not readily accessible by membrane lipids. Thus, we hypothesized that the TMD of STING might function as a potential ion channel.

We have previously reported the compound C53 as an unusual STING agonist (Lu et al, 2022). C53 binds precisely to the lower part of STING-TMD and appears to fully block the channel (Fig. 1C). Thus, we tested C53 as a potential inhibitor of STING's ion channel function and examined its impact on STING-mediated TBK1 activation as well as LC3 lipidation. Remarkably, the presence of C53 fully abolished STING-dependent LC3 lipidation without affecting TBK1 activation (Fig. EV1A), suggesting that STING-mediated noncanonical autophagy is likely triggered through an ion channel function of STING. Although C53 was later found to disrupt STING trafficking (see below), this initial experiment directed our study to STING-mediated LC3 lipidation instead of TBK1 signaling.

## Activated STING traffics through the Golgi complex

To explore how the potential STING channel could trigger LC3 lipidation, we first investigated where STING might induce LC3

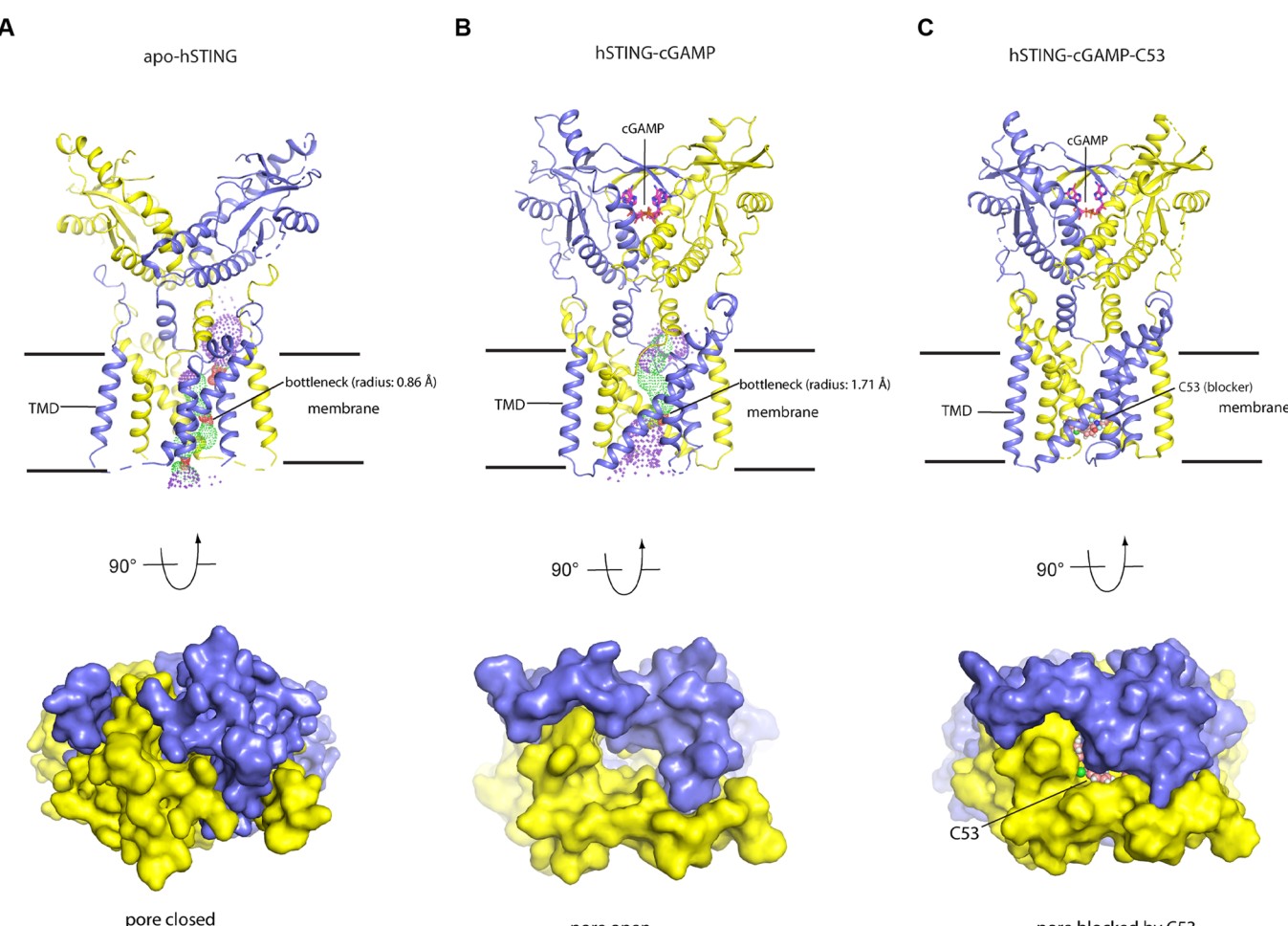

**Figure 1. Structural analysis of pore formation in the transmembrane domain of ligandbound and apo STING.**

The protomers of STING were shown in cartoon with blue and yellow colors. The cGAMP and C53 were displayed with stick and sphere mode, respectively. The pore radii (spheres) were calculated using HOLE. (A) Upper panel, the TMD of human apo-STING (PDB: 6NT5) forms a pore with a radius at its constriction point, its narrowest neck, measuring 0.86 Å; lower panel, the bottom view of pore with surface show. (B) Upper panel, the TMD of human cGAMP-bound STING (PDB: 8IK3) forms a pore with a radius of 1.71 Å at its constriction point; lower panel, the bottom view of pore with surface show. (C) Upper panel, the pore of activated STING (PDB: 7SII) is blocked by compound C53; lower panel, the bottom view of the pore with surface show.

lipidation in cells. Multiple subcellular localizations have been previously described as active sites for the STING complex, including ER-Golgi-intermediate compartments (ERGIC) (Dobbs et al, 2015; Gui et al, 2019), the Golgi body (Fang et al, 2021; Kong et al, 2023; Mukai et al, 2016), and post-Golgi vesicles (Fang et al, 2021; Fischer et al, 2020; Saitoh et al, 2009). We first characterized time-dependent STING trafficking upon ligand stimulation. U2OS cells are widely used for immunofluorescence studies, but no STING was expressed in this cell line. We established a monoclonal U2OS cell line stably expressing low levels of human STING. This cell line showed robust STING trafficking comparable to endogenous STING in human BJ and HT1080 cells, but is a better model to study STING trafficking as it has consistent STING trafficking throughout the cell population, allowing for careful dissection of the trafficking steps upon ligand stimulation.

Upon cGAMP delivery into U2OS cells, STING showed dramatic subcellular trafficking and strongly colocalized with the cis- and trans-Golgi markers, GM130 and Golgin97, respectively, about 30 min after cGAMP exposure (Fig. 2A–F). Remarkably, the occupation of the Golgi membrane by STING caused striking morphological changes of the Golgi complex, as shown by more extensive overlap between GM130 and Golgin97 (Fig. EV1B–D) as well as swelled Golgi stacks at 30 min (Figs. 2A,D and EV1B,E). After this, a large number of STING vesicles formed out of the Golgi complex (Fig. 2A–F), and the Golgi markers returned to resting morphology (Fig. EV1B–E). Despite the extensive clustering of Golgi-derived STING vesicles that appeared to partially overlap with Golgi markers, these STING vesicles are apparently different membranes outside of the Golgi stacks (Fig. 2A,D).

STING trafficking caused robust relocalization of the trans-Golgi network (TGN) marker TGN46 (also known as TGN38 or Trans-Golgi Network Protein 2, TGOLN2). While TGN46 showed extensive overlap with Golgin97 in resting conditions, the arrival of STING to the Golgi 30 min after cGAMP binding was accompanied by the budding of TGN46, but not Golgin97, from the trans-Golgi stack (Fig. 2G–I). The TGN46 vesicles and Golgi-derived STING vesicles appeared to be distinct with only partial overlap (Figs. 2J and EV1F).

STING trafficking through the Golgi complex was also observed when hSTING was activated by diABZI (Fig. EV1G,H), a non-nucleotide-based synthetic STING agonist (Ramanjulu et al, 2018). Similarly, endogenous STING in HT1080 and BJ cells were both found to traffic through the Golgi upon cGAMP treatment (Fig. EV2A,B). Mouse STING (mSTING) stably expressed in U2OS also trafficked through the Golgi upon cGAMP delivery (Fig. EV2C–H). Thus, in different cell lines, ligand binding activates robust STING trafficking through the Golgi complex to post-Golgi vesicles (Fig. 2K).

### STING-induced LC3 puncta primarily localize to post-Golgi, endosome-like vesicles

Since STING showed robust trafficking through the Golgi stacks, we asked whether STING stimulated LC3 lipidation directly on the Golgi membrane or post-Golgi vesicles. The cGAMP-stimulated endogenous LC3 puncta had little overlap with the Golgi (Figs. 3A–C and EV3A–C). The LC3 puncta did not appear 30 min after cGAMP exposure when most STING accumulated at the Golgi (Figs. 3A and EV3A,C), and their subsequent occurrence

correlated with STING trafficking to post-Golgi vesicles (Fig. 2A–F). The morphology of the triggered LC3 puncta resembled post-Golgi clusters of STING vesicles (Figs. 2A,D, 3A, and EV3A,C). Indeed, three-channel imaging showed that these LC3 puncta largely colocalized with STING but not Golgin97 (Fig. EV3D,E). Thus, these data strongly support that STING stimulates LC3 lipidation primarily on post-Golgi vesicles.

These observations are consistent with recent findings that cGAMP stimulates LC3 lipidation on single membrane STING vesicles (Fischer et al, 2020). To further explore the identity of these vesicles, we co-stained LC3 with different organelle markers at different time points after cGAMP treatment. This revealed a significant colocalization between the triggered LC3 puncta with the early endosome marker RAB5, but not Golgin97 (Figs. 3D,E and EV3C). Of note, the total punctate intensities of RAB5 substantially increased after STING left the Golgi (Fig. 3D), suggesting that post-Golgi STING vesicles likely developed endosome-like properties and recruited RAB5. Consistently, a sharp increase in STING colocalization with RAB5 was also observed in its post-Golgi stage (Fig. 3F–H). Further imaging revealed that cGAMP-induced LC3 puncta were typically found on STING vesicles positive for RAB5 (Fig. EV3F). Interestingly, STING and LC3 puncta also appeared to overlap with the recycling endosome marker RAB11 (Fig. 3I–K), consistent with a recent study that STING trafficking to lysosomes for degradation occurred through RAB11-positive vesicles (Kuchitsu et al, 2023). STING also developed increased colocalization with the late endosome/lysosome markers CD63 and LAMP1, but to a much lower extent compared with RAB5 (Fig. EV3G–L), reflecting partial late endosome properties of STING vesicles or STING delivery to the lysosome for degradation. Together, these results suggest a gradual acquisition of endolysosomal properties by post-Golgi STING vesicles where most LC3 puncta are detected.

### STING deacidifies post-Golgi vesicles

Our data suggest that STING triggered LC3 lipidation mostly on post-Golgi, endosome-like vesicles. Given that C53, a chemical that binds to the transmembrane pore of STING, selectively blocks STING-mediated LC3 lipidation without affecting TBK1 activation or subsequent STING phosphorylation (Fig. EV1A), we hypothesized that a potential ion channel function of STING triggers LC3 lipidation on post-Golgi STING vesicles. STING-mediated LC3 lipidation is known to depend on the V-ATPase proton pump that directly recruits ATG16L1 (Fischer et al, 2020). In other contexts, the same V-ATPase-ATG16L1 axis senses the deacidification of peripheral organelles such as phagosomes and endolysosomes and triggers direct LC3 lipidation onto these compartments, known as CASM (Cross et al, 2023; Durgan and Florey, 2022; Wang et al, 2022). Since STING also activates CASM through V-ATPase, we examined whether STING activation could raise the pH of its post-Golgi trafficking vesicles.

Lyso-pHluorin is a pH sensor of endolysosomes, the fluorescence of which is quenched at acidic pH and activated upon pH neutralization or endolysosomal deacidification (Rost et al, 2015), as validated in Fig. EV4A as well as in our recent study of lysosomal repair (Tan and Finkel, 2022). We tested if lyso-pHluorin could serve as a pH sensor of STING vesicles which showed endolysosomal properties (Fig. 3). Remarkably, STING activation triggered

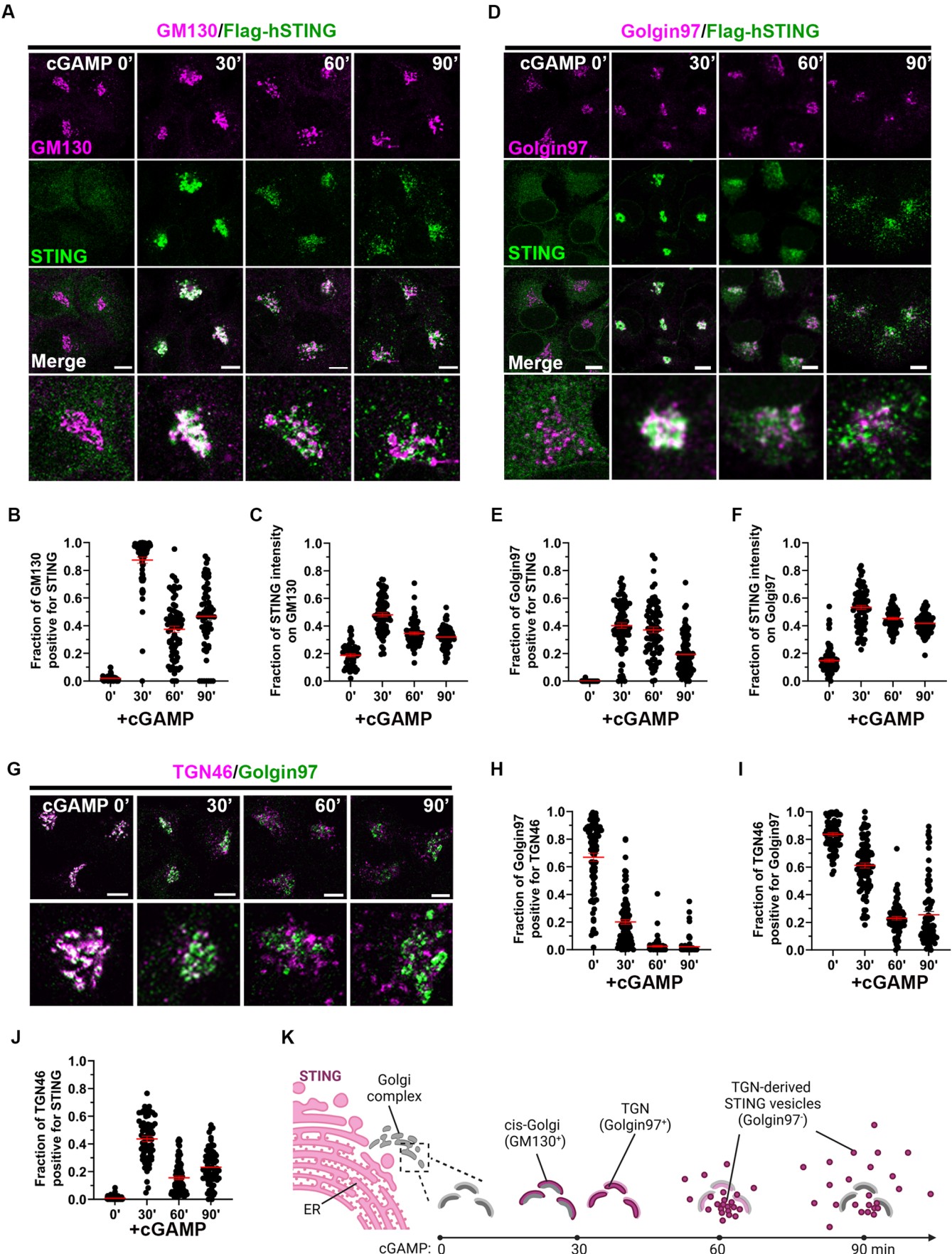

Figure 2. Activated STING traffics through the Golgi complex.

(A) STING traffics through the *cis*-Golgi upon cGAMP binding. Monoclonal U2OS cells stably expressing Flag-tagged human STING (hSTING) were stimulated with 1 μM cGAMP and fixed at indicated time points for the co-staining of STING and the cis-Golgi marker GM130. (B, C) Quantification of the colocalization between STING and GM130 in (A). Mean ± SEM; $n = 53$, 80, 79, and 67 random cells for 0′, 30′, 60′, and 90′, respectively. (D) STING traffics through the *trans*-Golgi upon cGAMP binding. U2OS Flag-hSTING cells the same as in (A) were stimulated with 1 μM cGAMP and fixed at indicated time points for the co-staining of STING and the trans-Golgi marker Golgin97. (E, F) Quantification of the colocalization between STING and Golgin97 in (D). Mean ± SEM; $n = 81$, 89, 82, and 88 random cells for 0′, 30′, 60′, and 90′, respectively. (G) STING trafficking triggers TGN46 budding from the *trans*-Golgi. U2OS Flag-hSTING cells the same as in (A) were stimulated with 1 μM cGAMP and fixed at indicated time points for the co-staining of two different TGN markers TGN46 and Golgin97. (H, I) Quantification of the colocalization between TGN46 and Golgin97 in (G). Mean ± SEM; $n = 83$, 100, 77, and 90 random cells for 0′, 30′, 60′, and 90′, respectively. (J) Quantification of the colocalization between TGN46 and STING in U2OS Flag-hSTING cells. Mean ± SEM; $n = 80$, 66, 80, and 70 random cells for 0′, 30′, 60′, and 90′, respectively. (K) Schematic illustration of STING trafficking through the Golgi stacks to post-Golgi vesicles. Created using Biorender. Data Information: Bar, 10 μm for all cell imaging panels. Statistical significance was determined by unpaired, two-tailed $t$ tests for all quantifications. Source data are available online for this figure.

the formation of lyso-pHluorin puncta in both U2OS cells reconstituted with human STING (hSTING) and BJ cells expressing endogenous hSTING (Figs. 4A,B and EV4B). Similar lyso-pHluorin puncta were observed when mouse STING was expressed in U2OS cells and activated by DMXAA, a chemical agonist selectively activating mouse STING (Conlon et al, 2013; Gao et al, 2013; Prantner et al, 2012) (Fig. 4C,D). The activation of neither human nor mouse STING caused any puncta formation of EGFP-galectin3 (Fig. EV4C), a sensor of endolysosomal membrane damage (Maejima et al, 2013; Skowyra et al, 2018), suggesting that the membrane integrity of STING vesicles was not compromised and that the observed pH neutralization was likely due to the altered activity of potential ion channels. The vesicle deacidification activity of STING appeared to be highly conserved, given that lyso-pHluorin puncta were equally detected when *Xenopus tropicalis* STING (XtSTING) was activated by cGAMP in U2OS cells (Fig. 4E,F). We consistently noticed the development of brighter, STING-dependent lyso-pHluorin puncta over time. This was confirmed using time-lapse live-cell imaging that showed substantial lyso-pHluorin puncta 60–120 min after the activation of mSTING (Fig. 4G,H), consistent with STING trafficking to endosome-like vesicles at this time (Fig. 3). Thus, vesicle deacidification by STING might be sensed by the V-ATPase proton pump for direct LC3 lipidation on these vesicles. Consistent with vesicle deacidification being upstream of LC3 lipidation, STING-induced lyso-pHluorin puncta formation was normal in ATG5- and ATG7-knockout cells (Fig. EV4D).

To further characterize whether the observed lyso-pHluorin puncta represent STING vesicles or other nearby, pre-existing endolysosomes, we developed a protocol to stain for STING in the same cells after the induction of lyso-pHluorin puncta by cGAMP (Fig. EV4E). This approach revealed extensive colocalization of cGAMP-induced lyso-pHluorin puncta with STING (Fig. 4I,J) as well as RAB5 (Fig. 4K). It is surprising that Lyso-pHluorin, a pH sensor of late endosome/lysosomes (Rost et al, 2015; Tan and Finkel, 2022), detected STING-dependent vesicle deacidification on RAB5-positive STING vesicles (Fig. 4K). To further examine such deacidification events, we investigated the relative localization of STING-induced Lyso-pHluorin puncta and different organelle markers in live cells. Consistent with STING colocalization with RAB5 and RAB11 (Fig. 3F–K), cGAMP-induced Lyso-pHluorin puncta also overlapped with both RAB5 and RAB11, but with apparently less association with various late endosome/lysosome markers such as RAB7 and CD63 (Fig. EV4F, quantified in Fig. 4L). These results suggest that STING-induced Lyso-pHluorin puncta are localized to STING vesicles different from late endosome/

lysosomes. We added an mCherry tag to the C-terminal end of the Lyso-pHluorin probe so that we could track both the localization and pH-dependent (de)quenching of the probe. This probe again confirmed that STING-induced Lyso-pHluorin puncta represented a fraction of the probe with a weaker mCherry signal different from the pre-existing bright mCherry puncta on late endosome/lysosomes (Fig. EV4G). Together, these data indicate that post-Golgi endosome-like vesicles are deacidified by STING, consistent with V-ATPase-ATG16L1-meidated LC3 lipidation.

## STING mediates proton flux in vitro

The vesicle deacidification activity of STING further suggests that the transmembrane pore of STING might function as an ion channel allowing protons and/or other ions to pass through the membrane, which is consistent with our structural analysis (Fig. 1). To test this hypothesis, we conducted an in vitro ion flux assay (Su et al, 2016) using purified STING reconstituted onto proteoliposomes (Fig. 5A). In this assay, proton influx was driven by a strong gradient for the efflux of potassium across the membrane. The proton-sensitive dye 9-Amino-6-chloro-2-methoxyacridine (ACMA) was used to monitor the pH change inside the vesicles. A potassium ionophore valinomycin was added to initiate the flux. Surprisingly, in this system, STING showed constitutive activity in proton flux, which was not further enhanced by the addition of diABZI, a membrane-permeable STING agonist expected to activate STING from the luminal side of the liposomes (Fig. 5B). Nevertheless, the basal proton flux activity was strongly inhibited by C53 (Fig. 5B) which binds to and blocks the transmembrane pore of STING (Fig. 1C). Taken together, our structure analysis, cellular imaging, and in vitro ion flux assay suggest that STING functions as a proton channel to deacidify at least Golgi-derived vesicles, consistent with cGAMP-induced STING trafficking to perinuclear acidic compartments.

STING-mediated ion flux is likely selective for protons, unique ions with an extremely small size. First, the heavily positively charged cytosolic side of this channel might not allow the efflux of cation ions such as Na+ and K+ (Appendix Fig. S1A). Second, the relatively small size of the pore might preclude the passage of anion ions like hydrated Cl- with an estimated radius of 3.32 Å. Finally, mutations of various amino acid residues in the center of the transmembrane pore in human STING (Appendix Fig. S1B,C) failed to identify any mutants completely dead in vesicle deacidification (Appendix Fig. S1D,E), among which those triggered weaker or no lyso-pHluorin puncta turned out to be unstable (Appendix Fig. S1F). One mutant L54E induced brighter

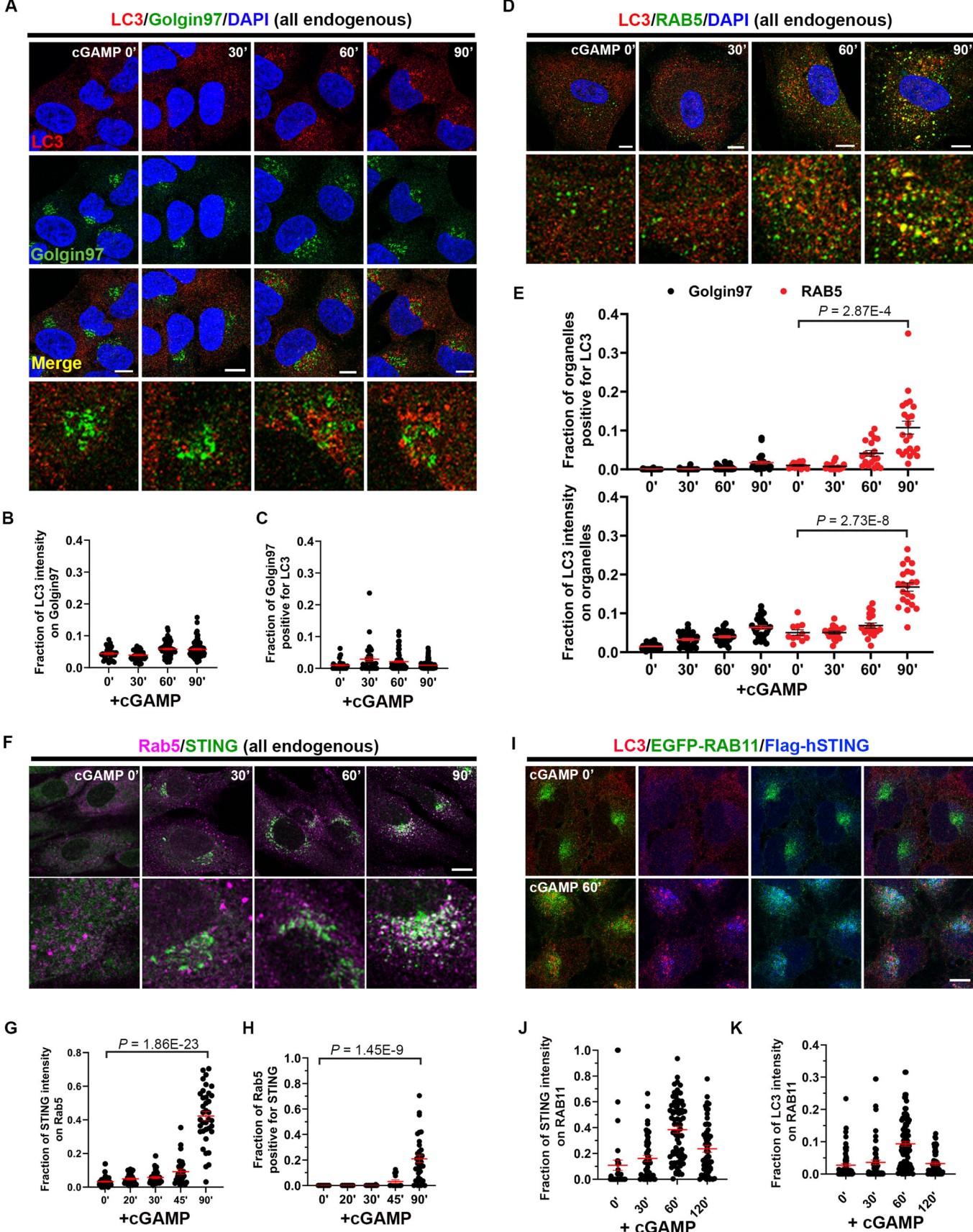

◄ **Figure 3. STING induces LC3 puncta on post-Golgi, endosome-like vesicles.**

(**A**) STING induces LC3 puncta outside of the Golgi. Monoclonal U2OS Flag-hSTING cells were stimulated with 1 µM cGAMP and fixed at indicated time points for the co-staining of endogenous LC3 and the *trans*-Golgi marker Golgin97. (**B, C**) Quantification of the colocalization between LC3 and Golgin97 in (**A**). Mean ± SEM; *n* = 31, 40, 72, and 92 random cells for 0', 30', 60', and 90', respectively. (**D**) STING induces LC3 puncta colocalize with RAB5. BJ cells were stimulated with 1 µM cGAMP and fixed at indicated time points for the co-staining of endogenous LC3 and the RAB5. (**E**) Quantification of the colocalization between LC3 and RAB5 in (**D**) or Golgin97 in Fig. EV3C. Mean ± SEM; *n* = 31, 30, 25, 27, 11, 18, 19, and 22 random cells from left to right. (**F**) Post-Golgi STING puncta colocalize with RAB5. BJ cells were stimulated with 1 µM cGAMP and fixed at indicated time points for the co-staining of endogenous STING and RAB5. (**G, H**) Quantification of the colocalization between STING and RAB5 in (**G**). Mean ± SEM; *n* = 39, 27, 35, 31, 37, 35, 15, 21, 13, and 34 random cells from left to right. (**I**) Co-staining of Flag-hSTING and LC3 in EGFP-RAB11 cells. U2OS cells stably expressing hSTING and EGFP-RAB11 were stimulated with cGAMP and fixed at indicated time points for the staining of endogenous LC3 and Flag-tagged hSTING. (**J, K**) Quantification of the fraction of STING intensity or LC3 intensity on RAB11 in (**I**). Mean ± SEM; *n* = 38, 53, 72, and 55 random cells from left to right. Data Information: Bar, 10 µm for all cell imaging panels. Statistical significance was determined by unpaired, two-tailed *t* tests for all quantifications. Source data are available online for this figure.

lyso-pHluorin puncta than wild type and any other mutants of STING (Appendix Fig. S1C–E), suggesting increased proton efflux by this mutant. As the L54E mutation is located near the narrowest neck of the channel (Appendix Fig. S1G), the increased proton efflux is likely mediated by an interaction between protons and E54 that allows more efficient proton passage through the channel.

## The channel of STING mediates vesicle deacidification and LC3 lipidation

The data strongly argue for an ion channel function of STING that deacidifies its post-Golgi trafficking vesicles, thus allowing for the activation of the V-ATPase-ATG16L1 axis for LC3 lipidation. Since we were not able to find a STING mutant with a selective loss-of-function in vesicle deacidification (Appendix Fig. S1C–E), we next focused on the STING channel blocker C53 in characterizing the role of ion flux in vesicle deacidification and LC3 lipidation.

For C53 to be a valid blocker of the ion channel function of STING, we first need to rule out its potential impact on STING trafficking from the ER to post-Golgi vesicles. We have recently shown that although C53 binds to the transmembrane pore of human STING, it also triggers STING oligomerization and TBK1 activation (Lu et al, 2022). Surprisingly, C53 alone induced large numbers of STING puncta with little colocalization with the Golgi (Appendix Fig. S2A). The addition of C53 together with either cGAMP or diABZI prevented STING trafficking to the Golgi, causing the accumulation of both endogenous and ectopic STING in punctate structures, similar to what was stimulated by C53 alone (Fig. 6A–C; Appendix Fig. S2A–D). Thus, the presence of C53 redirects STING to an abnormal route that blocks STING trafficking at a pre-Golgi stage.

To assess the impact of C53 without blocking STING trafficking, we tested C53 addition after STING arrived at the Golgi. In U2OS Flag-STING monoclonal cells, most STING accumulated at the Golgi 30 min after cGAMP treatment (Figs. 2 and 6C,D). Adding C53 at this time point did not block STING trafficking to post-Golgi vesicles (Fig. 6C,D; Appendix Fig. S2C,E). However, C53 still almost completely blocked STING-dependent vesicle deacidification (Fig. 6E,F) as well as LC3 lipidation, without affecting TBK1 activation (Fig. 6G–J). Compared with cGAMP, diABZI triggered slower STING trafficking, with most STING found at the Golgi 60 min after stimulation (Fig. EV1G,H). Adding C53 at this time still fully blocked diABZI-induced LC3 lipidation (Appendix Fig. S2F).

We further tested adding C53 after STING fully arrived at post-Golgi vesicles. Specifically, C53 was added 2 h after cGAMP or

diABZI when vesicle deacidification was already detected. Under these conditions, STING-mediated LC3 lipidation was not reversed by C53 as it was already saturated by 2 h (Appendix Fig. S2G). Remarkably, C53 still dramatically suppressed STING-mediated vesicle deacidification, and even fully cleared diABZI-induced lyso-pHluorin puncta (Figs. 7A–C and EV5A). Thus, the fraction of Lyso-pHluorin probe on endosome-like, post-Golgi STING vesicles (Fig. EV4G) can only be detected when these vesicles are deacidified by STING. Taken together, when C53 was used appropriately, it selectively suppresses STING-mediated vesicle deacidification and LC3 lipidation, without affecting STING trafficking or TBK1 signaling.

## STING-mediated cell death depends on its transmembrane channel

Overactivation of STING is known to cause cell death through an interferon-independent function of STING (Tang et al, 2016; Wu et al, 2019; Wu et al, 2022). A C-terminal "RHLR" motif of STING required for its trafficking from the ER is critical for STING-dependent cell death (Gui et al, 2019; Wu et al, 2019). We recapitulated STING-mediated cell death in U2OS cells stably expressing hSTING and examined whether it required the ion channel function of STING (Fig. 7D). Cells were briefly treated with cGAMP/digitonin buffer for 10 min and then changed back to original media. Extensive cell death was observed 24 h after cGAMP treatment (Fig. 7D,E). Strikingly, C53 addition 2 h after cGAMP treatment fully blocked cGAMP-induced cell death (Fig. 7D,E). It is of note that C53 alone, which is sufficient to activate TBK1 signaling for interferon production, did not cause any cell death in these tests (Fig. 7E). The L54E mutant of STING, which showed increased vesicle deacidification activity (Appendix Fig. S1C–E), consistently triggered more cell death than wild-type STING expressed in similar levels compared with the mutant (Fig. 7F,G). Thus, STING-mediated cell death under these conditions is dependent on its ion channel function but independent of TBK1 signaling.

STING-dependent cell death appears to involve multiple mechanisms including ferroptosis (Tang et al, 2016; Wu et al, 2019; Wu et al, 2022). Two ferroptosis inhibitors both partially suppressed, but failed to fully block, STING-dependent cell death (Fig. EV5B), suggesting the presence of additional cell death mechanisms. Since both STING-dependent LC3 lipidation and cell death relied on the STING channel, we asked if LC3 lipidation was required for STING-dependent cell death. Overexpression of a bacterial effector SopF (Fischer et al, 2020; Xu et al, 2019) strongly

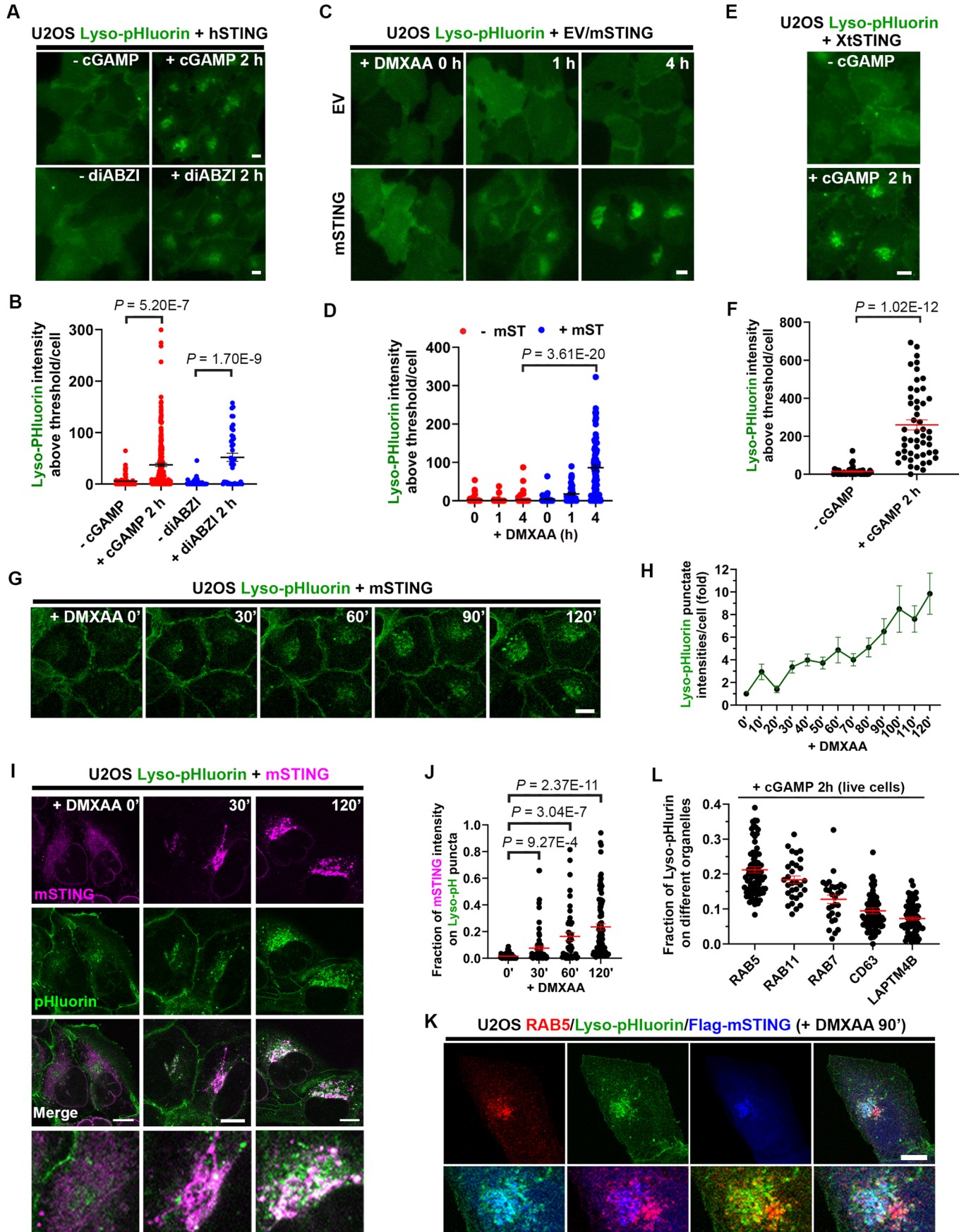

### Figure 4.   STING deacidifies its post-Golgi trafficking vesicles.

(A) Activation of human STING (hSTING) triggers lyso-pHluorin puncta. U2OS cells stably expressing hSTING and a genetically encoded pH sensor lyso-pHluorin were stimulated as indicated and the lyso-pHluorin puncta were monitored by wide-field live-cell imaging. (B) Quantification of the average intensities of lyso-pHluorin puncta in (A). Mean ± SEM; $n = 73, 249, 50$, and 42 random cells from left to right. (C) Activation of mouse STING (mSTING) triggers lyso-pHluorin puncta. U2OS cells stably expressing mSTING and lyso-pHluorin were stimulated with DMXAA and the lyso-pHluorin puncta were monitored by wide-field live-cell imaging. (D) Quantification of the average intensities of lyso-pHluorin puncta in (C). Mean ± SEM; $n = 70, 44, 83, 44, 53, 89$ random cells from left to right. (E) Activation of Xenopus tropicalis STING (XtSTING) triggers lyso-pHluorin puncta. U2OS cells stably expressing XtSTING and lyso-pHluorin were stimulated with cGAMP and the lyso-pHluorin puncta were monitored by wide-field live-cell imaging. (F) Quantification of the average intensities of lyso-pHluorin puncta in (E). Mean ± SEM; $n = 50$ (−cGAMP), 42 ( + cGAMP 2 h) random cells. (G) Time-lapse confocal live-cell imaging shows mSTING-induced lyso-pHluorin puncta upon DMXAA treatment. (H) Quantification of the average intensities of lyso-pHluorin puncta in (G). Mean ± SEM; $n = 32$ cells. (I) Confocal images showing STING-induced lyso-pHluorin puncta overlapping with STING. U2OS cells stably expressing Flag-mSTING and lyso-pHluorin were stimulated with DMXAA and then fixed at indicated time points for the staining of mSTING by Flag antibody. See also Fig. EV4E. (J) Quantification of the colocalization between STING and lyso-pHluorin in (I). Mean ± SEM; $n = 55, 76, 47$, and 93 random cells from left to right. (K) Confocal images showing Lyso-pHluorin puncta on STING vesicles positive for RAB5. U2OS cells stably expressing Flag-mSTING and lyso-pHluorin were fixed 90 min after DMXAA stimulation for immunostaining of STING and RAB5. (L) Quantification of the colocalization of lyso-pHluorin with early/late endosome markers in live-cell imaging. Mean ± SEM; $n = 83, 29, 30, 81$ and 86 random cells from left to right. See images in Fig. EV4F. Data Information: Bar, 10 µm for all cell imaging panels. Statistical significance was determined by unpaired, two-tailed $t$ tests for all quantifications. Source data are available online for this figure.

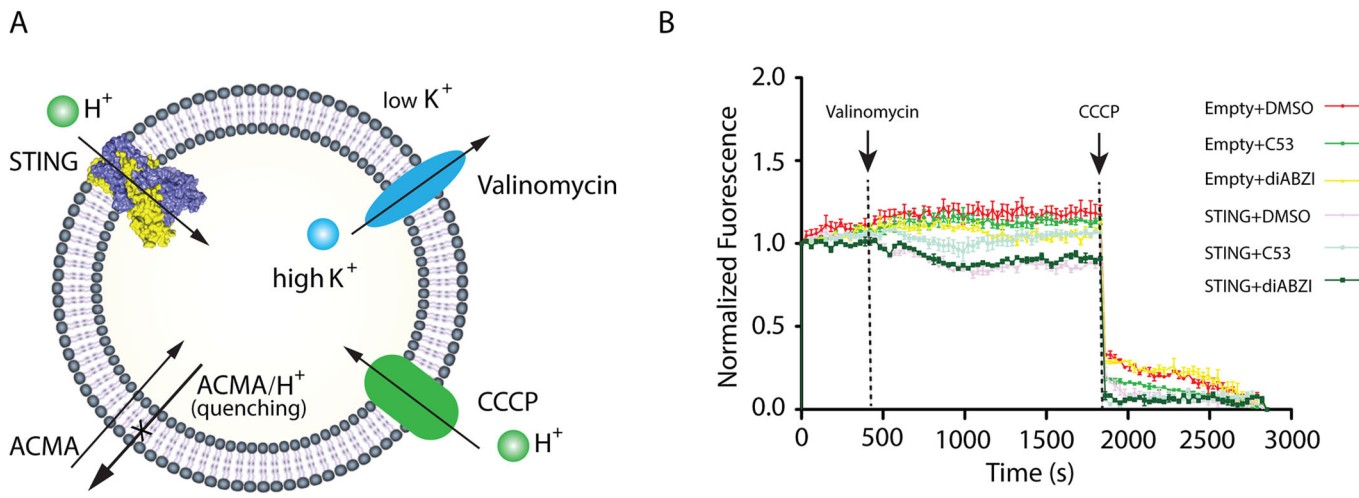

**A**

**B**

### Figure 5.   In vitro analysis of human STING as a proton channel.

(A) Schematic illustration of the fluorescence-based proton flux assay driven by a $K^+$ gradient. Vesicles were loaded with 450 mM KCl and then diluted into flux buffer with 450 mM NaCl in the presence of ACMA. Valinomycin was added to initiate the flux and proton-ionophore carbonyl cyanide chlorophenylhydrazone (CCCP) was used to collapse the electrical gradient. (B) Human STING-containing vesicles (1:100 wt/wt protein-to-lipid ratios) were diluted into buffer containing 450 mM NaCl. Valinomycin and CCCP were added at ~300 s and 1800 s, respectively (arrowhead and gray dashed line) and fluorescence was detected over time. Empty (without STING protein) vesicles were used as control. Mean ± SEM; $n = 6$ independent experiments. Source data are available online for this figure.

suppressed STING-mediated LC3 lipidation (Fig. 7H,I), but did not cause any defects in cell death (Fig. 7J). Similarly, the deletion of ATG7 also blocked LC3 lipidation but accelerated STING-dependent cell death (Fig. EV5C–E). Thus, STING-dependent LC3 lipidation appears to antagonize STING-mediated cell death.

In summary, we discovered an unexpected ion channel function of STING that allows the efflux of protons from post-Golgi STING vesicles, leading to vesicle deacidification, noncanonical autophagy, and cell death, all independent of STING-induced TBK1/interferon signaling (Fig. 7K). STING-mediated vesicle deacidification can be captured by lyso-pHluorin, a genetically encoded lysosomal pH sensor that is also targeted to post-Golgi STING vesicles. Although the channel blocker C53 disrupts STING trafficking, its addition after STING arrives at the Golgi or post-Golgi vesicles can selectively block the ion channel function of STING. The new function of STING appears to be highly conserved as STING from humans, mice, and frogs all stimulated vesicle deacidification when

activated (Fig. 4A–F). Such changes in vesicle pH likely activate the V-ATPase-ATG16L1 axis for LC3 lipidation onto STING vesicles (Fischer et al, 2020), resembling other CASM inducers (Cross et al, 2023; Tan and Finkel, 2022; Wang et al, 2022; Xu et al, 2019).

## Discussion

How STING triggers noncanonical autophagy and cell death are fundamental questions in innate immunity. In this study, we found that activated STING forms a transmembrane channel that mediates both noncanonical autophagy and cell death. The channel of STING allows the efflux of protons, leading to the deacidification of post-Golgi STING vesicles. STING-mediated vesicle deacidification mimics many other cellular stresses such as lysosomal membrane damage or exposure to proton ionophores, instigating the same downstream effectors for the induction of noncanonical

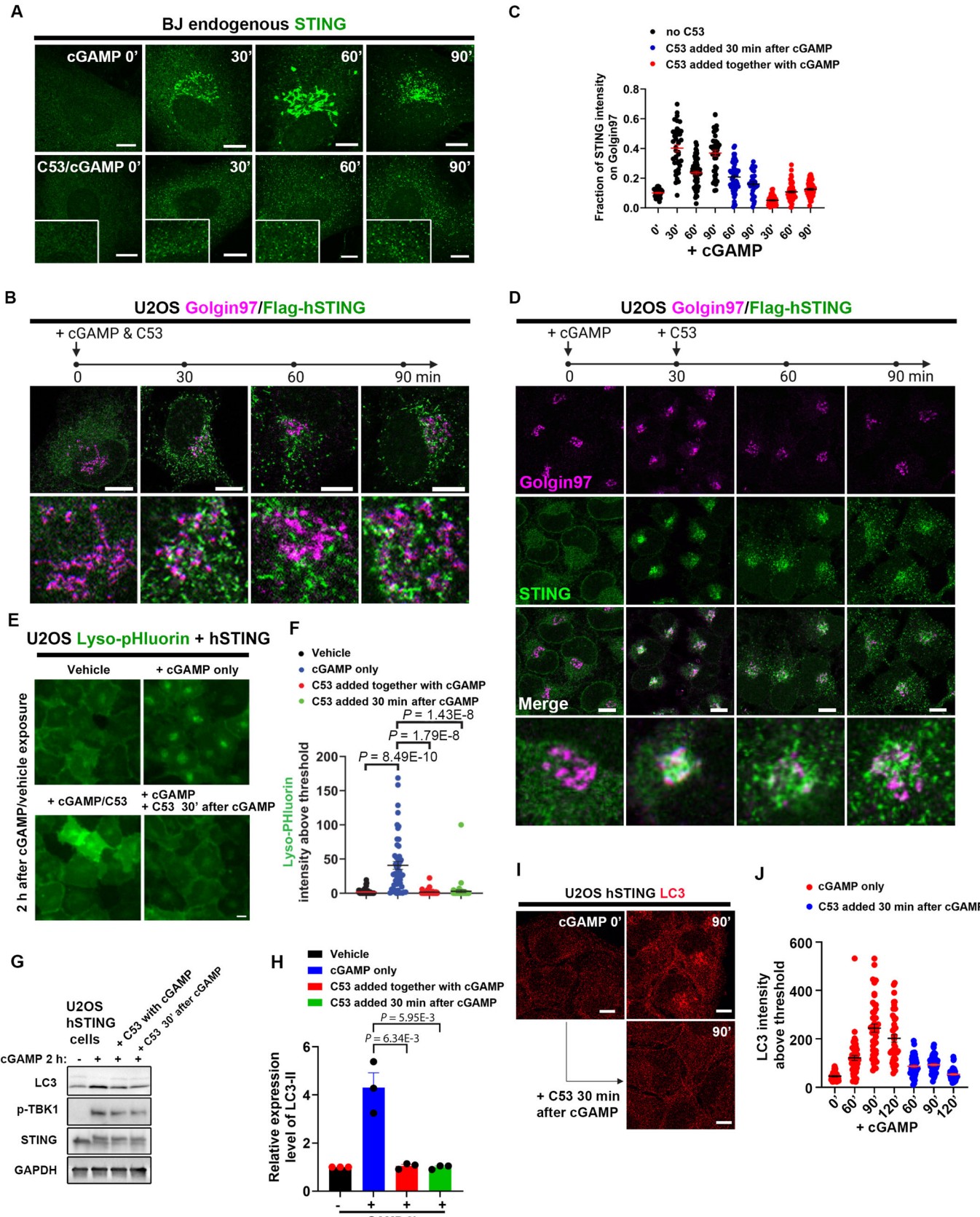

**Figure 6. The STING channel is required for vesicle deacidification and LC3 lipidation.**

(A) Compound C53, which binds to the transmembrane pore of STING, fully blocks the trafficking of endogenous STING from the ER to the Golgi. BJ cells were stimulated with cGAMP alone or cGAMP + C53 for indicated time periods, followed by fixation and immunostaining of STING. (B) Compound C53 blocks Flag-hSTING trafficking to the Golgi. Monoclonal U2OS Flag-hSTING cells were stimulated with cGAMP together with C53 for indicated time periods and then fixed for the immunostaining of STING and Golgin97. (C) Quantification of the colocalization between STING and Golgin97 in different conditions. Mean ± SEM; $n$ = 28, 39, 62, 44, 56, 32, 50, 64, and 56 random cells from left to right. Note that C53 addition 30 min after cGAMP does not block STING trafficking to or from the Golgi, but adding C53 together with cGAMP blocked STING trafficking to the Golgi. (D) C53 addition 30 min after cGAMP allows STING trafficking to the Golgi and post-Golgi vesicles. Monoclonal U2OS cells stably expressing hSTING were stimulated as indicated and then fixed for the immunostaining of STING and Golgin97. (E) C53 addition 30 min after cGAMP fully blocks STING-induced lyso-pHluorin puncta. U2OS cells stably expressing hSTING and lyso-pHluorin were stimulated as indicated, and the lyso-pHluorin puncta were monitored by live-cell imaging. (F) Quantification of the average intensities of lyso-pHluorin puncta in (E). Mean ± SEM; $n$ = 63, 57, 55, and 76 random cells from left to right. (G) C53 addition 30 min after cGAMP fully blocks STING-induced LC3 lipidation. Monoclonal U2OS Flag-hSTING cells were stimulated with cGAMP and C53 as indicated and all cells were harvested at 2 h for western blot analysis of indicated proteins. (H) Quantification of the relative expression level of LC3-II normalized to GAPDH from (G). Mean ± SEM; $n$ = 3. (I) C53 addition 30 min after cGAMP fully blocks STING-induced LC3 puncta formation. Monoclonal U2OS Flag-hSTING cells were stimulated with 1 µM cGAMP and fixed at indicated time points for the staining of endogenous LC3. (J) Quantification of the average intensities of LC3 puncta above threshold in (I). Mean ± SEM; $n$ = 38, 53, 46, 44, 53, 48, and 43 random cells for 0′, 30′, 60′, and 120′, respectively. Data Information: Bar, 10 µm for all cell imaging panels. Statistical significance was determined by unpaired, two-tailed $t$ tests for all quantifications. Source data are available online for this figure.

autophagy or CASM (Cross et al, 2023; Durgan and Florey, 2022; Tan and Finkel, 2022; Wang et al, 2022; Xu et al, 2019).

Our study suggests that the proton channel function of STING is highly conserved. A recent study identified a bacterial cGAMP receptor as a potential cGAMP-activated Cl⁻ channel (preprint: Tak et al, 2023). The surface charges on the cytosolic side of the pore suggest that STING might also allow for the flux of anions like Cl⁻ in addition to $H^+$. Alternatively, anion binding to the cytosolic side of the pore might allow for more efficient $H^+$ leakage. In addition, the size of the STING channel might change when it is transported to acidic post-Golgi vesicles, which in turn would affect the channel selectivity in ion flux. Thus, future studies are needed to thoroughly test the ion selectivity of the STING channel.

It has been controversial regarding the signaling compartments of STING. The extensive clustering of Golgi-derived STING vesicles near the microtubule-organization center (MTOC) has rendered it difficult to differentiate post-Golgi STING vesicles from the Golgi stacks which are themselves considered as part of the MTOC (Fig. EV5F, right). This situation is further complicated by the budding of certain TGN markers such as TGN46 during STING trafficking as well as by often overly simplified cartoon models of how key organelles involved in STING trafficking are organized in the cell (Fig. EV5F, left). Our in-depth analysis of ligand-stimulated STING trafficking in different cell lines strongly suggests the active sites of STING-mediated LC3 lipidation as post-Golgi vesicles. A fraction of lyso-pHluorin probe is localized to endosome-like STING vesicles separated from late endosome/lysosomes and is dequenched by STING-mediated vesicle deacidification. Although it seems likely that STING may neutralize the pH of the Golgi complex with a relatively low level of LC3 lipidation activity, most endogenous LC3 puncta were observed outside of the Golgi. Surprisingly, the presence of C53 blocked STING trafficking at a pre-Golgi stage, leading to STING accumulation in punctate structures that were still capable of activating TBK1 but not LC3 lipidation. The problem can be solved by adding C53 after STING traffics to the Golgi or post-Golgi vesicles, allowing selective inhibition of the ion channel functions of STING by C53.

During the preparation of this manuscript, a similar study by Liu et al was published (Liu et al, 2023a). The two studies both uncovered STING-mediated deacidification of perinuclear vesicles but through different approaches and pH sensors. Both studies observed STING-mediated proton flux in vitro and achieved a block of STING's ion channel function by C53 that tightly locks the transmembrane pore. Of note, purified STING was found to be constitutively active in proton flux, suggesting that purified STING might be partially activated when incorporated onto membranes where the autoinhibition through head-to-head binding is absent (Liu et al, 2023b). Both studies also observed a block of STING-dependent vesicle deacidification and cell death by C53, while our study established a more strict protocol in assessing the impact of C53 as a STING channel blocker without disrupting its trafficking, in addition to identifying post-Golgi vesicles as the major sites of action for the STING channel. It is of note that it is unclear how STING-dependent LC3 lipidation contributes to noncanonical autophagy and viral suppression.

While the cGAS/STING pathway is critical for innate immunity, abnormal activation of this pathway has been increasingly implicated in various human diseases including Parkinson's and Alzheimer's diseases, as well as normal aging (Gulen et al, 2023; Sliter et al, 2018; Udeochu et al, 2023). The ion channel function of STING described here opens an avenue for new therapeutic strategies to target this pathway. Selective manipulation of the inflammation signaling or the ion channel activity, depending on the contexts of specific diseases, might be considered for optimal therapeutic outcomes.

## Methods

### Cell culture, chemicals, and treatments

BJ, U2OS, and 293T cells were from ATCC and were authenticated through short tandem repeat (STR) profiling. All relevant authentication data are publicly available from ATCC. These cell lines differ in their growth rates and morphologies. All cell lines in this study were free of contaminations from other cell lines or mycoplasma. Cells were maintained with mycoplasma reagent and potential contaminations are regularly monitored through polymerase chain reaction (PCR) detection. All cells were cultured at 37 °C with 5% $CO_2$ in Dulbecco's modified Eagle's medium (DMEM) supplemented with 8% fetal bovine serum (FBS) and penicillin/streptomycin. Cyclic [G(2',5')pA(3',5')p] or cGAMP (#CT-CGMAP, ChemieTek) was dissolved in water and stored at −20 °C. The other two STING agonists diABZI (#S8796, Selleck) and C53 (#37354, Cayman) were dissolved in DMSO and stored in aliquots at −80 °C. Monensin (#16488, Cayman) was dissolved in ethanol and stored at −20 °C. cGAMP was delivered to cells using a mild digitonin buffer (50 mM HEPES pH 7.0, 100 mM

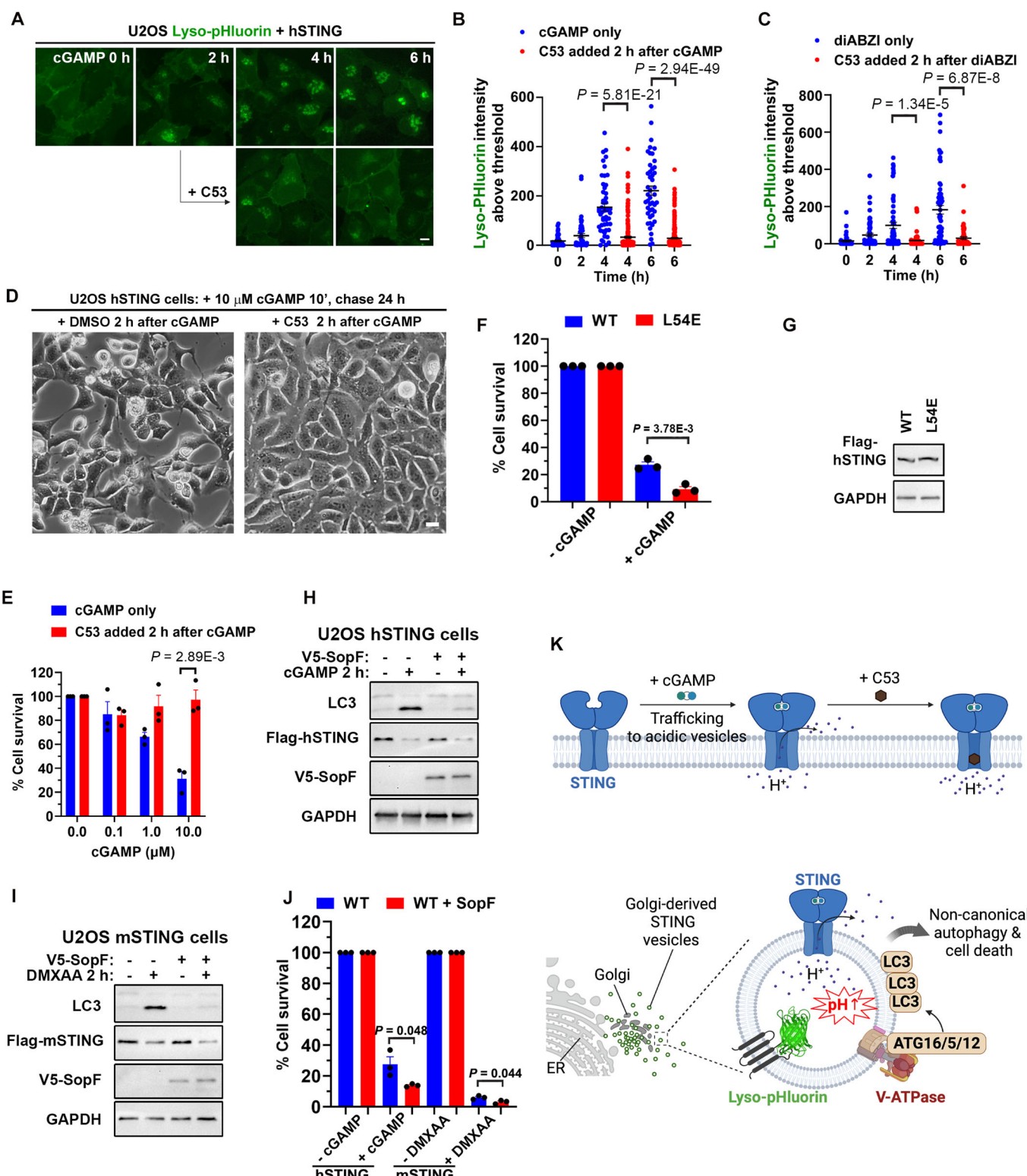

KCl, 3 mM MgCl₂, 0.1 mM DTT, 85 mM sucrose, 0.2% BSA, 1 mM ATP, 8 μg/mL digitonin) (Woodward et al, 2010). Cells were treated with the buffer containing 1 μM cGAMP for 5-10 min and then changed back to the original culture media. All other chemicals were directly added to the cell culture media.

## Antibodies

Antibodies for LAMP1 (sc-20011, IF 1:200); LAMP2 (sc-18822, IF 1:200); Golgin97 (sc-59820, IF 1:500); GAPDH (sc-365062, WB 1:5000); Tubulin (sc-5286, WB 1:3000) were from Santa Cruz

Figure 7.  Blocking the STING channel reverses vesicle deacidification and prevents STING-mediated cell death.

(A) C53 addition 2 h after cGAMP treatment strongly suppresses STING-induced lyso-pHluorin puncta. U2OS cells stably expressing hSTING and lyso-pHluorin were stimulated as indicated, and the lyso-pHluorin puncta were monitored by live-cell imaging. (B) Quantification of the average intensities of lyso-pHluorin puncta in (A). Mean ± SEM; $n = 50$, 51, 51, 191, 46, and 257 random cells from left to right. (C) Quantification of the average intensities of lyso-pHluorin puncta in Fig. EV5A. C53 addition 2 h after diABZI treatment strongly suppresses and reverses STING-induced lyso-pHluorin puncta. Cells were treated similarly as in (A) except that STING was activated by diABZI. Mean ± SEM; $n = 42$, 62, 65, 61, 61, and 50 random cells from left to right. (D) C53 addition 2 h after cGAMP treatment fully blocks STING-dependent cell death. Monoclonal U2OS Flag-hSTING cells were treated as indicated and the cell death was analyzed 24 h after cGAMP treatment. (E) Quantification of the cell death assays in (D). Percentages of cells compared with no cGAMP control were presented. Mean ± SEM. $n = 3$ independent experiments. (F) STING-L54E accelerates cell death. U2OS cells stably expressing Flag-hSTING WT or L54E were treated as indicated and cell death was analyzed 24 h after cGAMP treatment. (G) Western blot verifying that the expression levels of hSTING WT and L54E in (F) were similar. (H, I) Overexpression of SopF strongly blocks STING-induced LC3 lipidation. U2OS cells stably expressing hSTING (H)/mSTING (I) with or without SopF were treated as indicated and harvested 2 h after treatment for western blot. (J) Overexpression of SopF does not block STING-dependent cell death. U2OS cells stably expressing hSTING with or without SopF were treated as indicated and cell death was analyzed. Mean ± SEM; $n = 3$. (K) Schematic model of the ion channel function of STING in vesicle deacidification, noncanonical autophagy, and cell death. Top, cGAMP induces a transmembrane pore of STING as well as STING trafficking to post-Golgi acidic vesicles; compound C53 can be used to block the STING channel when added after STING trafficking to the Golgi or post-Golgi vesicles. Bottom, activated STING traffics through the Golgi complex to perinuclear clusters of endosome-like acidic vesicles; the channel function of STING causes proton efflux and vesicle deacidification, which can be sensed by a pH sensor lyso-pHluorin; the increased luminal pH of STING vesicles activates the V-ATPase-ATG16L1 axis for noncanonical LC3 lipidation onto STING vesicles; activation of the STING channel also triggers cell death. Created using Biorender. Data Information: Bar, 10 μm for all cell imaging panels. Statistical significance was determined by unpaired, two-tailed $t$ tests for all quantifications. Source data are available online for this figure.

Biotechnology. The GM130 antibody (610822, IF 1:1000) was from BD Biosciences. Flag (M2, IF 1:1000; F7425, WB 1:3000) antibody was from Sigma. LAMP1 rabbit mAb (#9091, IF 1:200); GM130 rabbit mAb (#12480, IF 1:300); Rab5 rabbit mAb (#3547, IF 1:200), Phospho-STING S366 (#50907, WB 1:1000); Phospho-TBK1 Ser172 rabbit mAb (#5483, WB/IF 1:1000) were from Cell Signaling. The following antibodies were from Proteintech: STING (19851-1-AP, WB 1:1000, IF 1:1000); LC3 (14600-1-AP, IF 1:1000, WB 1:1000, reacts with LC3A, LC3B, and LC3C); TGN46 (13573-1-AP, IF 1:1000); Golgin97 (12640-1-AP, IF 1:1000). Alexa-488/594- and Pacific Blue-conjugated secondary antibodies were obtained from ThermoFisher Scientific.

## DNA cloning

We used a lentiviral approach for stable protein expression. The DNA sequences of interest were PCR amplified and inserted into a lentiviral vector pCDH-CMV-MCS. When two genes are fused, a GSGSGS linker was used. Small epitope tags were added directly into PCR primers. To generate point mutations, two fragments of the target cDNA were amplified by PCR with their overlapping ends carrying the intended mutations. The two fragments were then fused together into the pCDH vector through infusion reactions. All new plasmids are verified by DNA sequencing.

## Stable cell line generation

All experiments in this study are based on stable lines. No transient transfection of DNA was used in any experiment to avoid DNA-induced activation of the cGAS/STING pathway. We used a lentiviral approach for stable protein expression. Lentiviruses were packaged in 293T cells by transfecting the pCDH vector carrying the gene of interest together with the packaging vectors pMD2.G and pSPAX2 using Lipofectamine 2000 (ThermoFisher). The culture media containing the viruses were collected 48 h after transfection, and were immediately used to infect target cells for stable protein expression. When needed the infected cells were selected by puromycin treatment. The lowest virus titer was used to achieve the desired infection rate. To generate knockout cell lines, CRISPR–Cas9 guide sequences were tested to identify the guide that effectively diminished the target protein in the CRISPR pool.

The lentiCRISPR v2 (Addgene, 52961) carrying the selected guide sequence was used for lentivirus packaging. The infection rate of U2OS cells by CRISPR viruses was remarkably high that no cell death was observed after puromycin treatment of infected pools when all uninfected control cells died. ATG7 CRISPR targeting sequence TATACAGTGTTCCAATAGCT was used when generating the ATG7-KO pools. ATG5-KO and ATG7-KO U2OS single clones were previously described (Tan and Finkel, 2022).

## Immunofluorescence

Cells were seeded on glass coverslips (Electron Microscopy Sciences, #7223001) in 24-well plates. After indicated cell stimulations, cells were fixed with 4% Paraformaldehyde (PFA) in phosphate-buffered saline (PBS) for 30 min at room temperature. Cells were then permeabilized with 0.1% Triton X-100 for 2 min, and blocked for 30 min with 1× fluorescent blocking buffer (ThermoFisher Scientific, #37565). The same buffer was further used for the dilution of primary and secondary antibodies in the following cell staining steps. Cells were incubated with primary antibodies at 4 °C overnight or at room temperature for 2 h. Unbound primary antibodies were washed away with PBS, and cells were further stained fluorescently labeled secondary antibodies. Cells were then washed more than three times to remove any unbound secondary antibodies. When 4′,6-diamidino-2-phenylindole (DAPI) staining was needed, cells were stained with DAPI for 3 min and then washed with PBS. The coverslips were mounted on slides using VECTASHIELD Mounting Medium (Vector Laboratories, #H-1700). Fluorescence images were collected using a Leica SP8 LIGHTNING confocal system with a built-in software Leica Application Suite X 3.5.5.19976. Different positive and negative controls were included to rule out nonspecific staining and any cross-talks between channels. Live-cell imaging was done using the same confocal system with an Okolabstage-top incubator. All images in the same panel were from the same experiment, followed with the same cell staining, image collection settings, and image processing.

## Immunoblotting

Cells with indicated treatments were briefly washed with cold PBS and then lysed with a lysis buffer containing 50 mM Tris-HCl, pH 7.5, 150 mM NaCl, 0.5% Triton X-100, 2 mM NaF, and a protease

inhibitor cocktail. The lysates were briefly sonicated to fully dissolve all membranes, followed by centrifugation at 15,000×g for 10 min. The supernatants were collected and heated at 95 °C for 5 min after mixing with equal volumes of 2× SDS loading buffer (0.1 M Tris-HCl, pH 6.8, 4% SDS, 20% glycerol, 2% 2-mercaptoethanol, 0.01% bromphenol blue). The protein samples were kept at −80 °C or directly moved to immunoblotting analysis. Protein electrophoresis was done using 4-20% precast polyacrylamide gel (Bio-Rad, #4561096), followed by protein transfer to 0.45-μm Nitrocellulose membranes using the Trans-Blot Turbo system. The membranes were then blocked with StartingBlock Blocking Buffer (ThermoFisher, #37542) for 30 min and then incubated sequentially with primary and secondary antibodies. After washing, the target proteins were detected Immobilon Forte Western HRP Substrate (Sigma-Aldrich, # WBLUF0500) in a ChemiDoc MP Imaging System.

## Lyso-pHluorin assay

Cells stably expressing Lyso-pHluorin at 50–70% confluency were treated with vehicle or indicated STING agonists to trigger STING trafficking and vesicle deacidification. For cGAMP treatment, the media from the cell culture dish were moved to a new tube and kept at 37 °C. Cells were treated with the digitonin/cGAMP buffer for 10 min, and the buffer was then replaced with the original warm culture media. Other chemicals were directly added to and mixed well in culture media. Cells were subsequently monitored for changes in their lyso-pHluorin signals every 30 min and live-cell images were collected at desired time points.

## Protein expression and purification

The human STING protein was expressed and purified following established protocols (Shang et al, 2019; Zhang et al, 2019). Briefly, The coding sequences of the human STING gene were inserted into a modified pEZT-BM vector, generating a STING fusion protein with a C-terminal Tsi3 tag (Lu et al, 2014). The plasmid was transfected into HEK293F cells using PEI with a mass ratio of 1:3 (plasmid : PEI) and the expression was enhanced by 3 mM sodium butyrate 12 h later. After 48 h of culturing, cells were collected and re-suspended into buffer A (20 mM Tris pH 7.5, 150 mM NaCl, 1 mM AEBSF and protease inhibitor cocktail). Then, cells were lysed by French press, and the membranes were collected using ultracentrifugation at 100,000×g. The membrane proteins were extracted using 1% DDM/CHS (5:1) and the insoluble part was removed by ultracentrifugation at 100,000×g. The supernatant was loaded onto the Tse3-conjugated resin pre-equilibrated with buffer B (20 mM Tris pH 8.0, 150 mM NaCl, 1 mM CaCl$_2$, 0.03% DDM and 0.003% CHS). The Tsi3 tag was removed using on-column digestion by adding 3C-protease overnight. The STING protein was further polished by applying to size exclusion chromatography (SEC) in buffer C (20 mM HEPES pH 7.5, 150 mM NaCl, 0.03% DDM, 0.003% CHS, and 1 mM TCEP).The peak was pooled, concentrated at 8 mg/ml and stored at −80 °C for flux assay.

## Fluorescence-based proton flux assay

Lipids of PC (20 mg/ml), PE (20 mg/ml) and PG (20 mg/ml) were mixed in chloroform at a 2:1:1 ratio and dried using a nitrogen stream, and trace mount of chloroform was removed using a vacuum chamber.

Dried lipids were suspended in internal buffer C (50 mM HEPES pH 7.4, 450 mM KCl and 5 mM DTT) generating the liposome with a concentration of 10 mg/ml. The unilamellar liposomes was formed by freeze–thawing cycle followed by sonication. An aliquot of liposomes were mixed with 0.7% DDM for 2 h at 25 °C. Human STING protein was then added to the liposome/DDM mixture (ratio 1:100) and incubated for 1 h at 25 °C. DDM was completely removed by adding Bio-beads SM2 (Bio-Rad) every 4 h at 4 °C for three times. Generated proteoliposome vesicles were collected and used for ion channel flux assays. Proteoliposomes (12 μl) were mixed with 160 μl external flux assay buffer D (5 mM HEPES pH 7.4, 450 mM NaCl, 5 mM DTT and 2 μM ACMA) in a 96-well fluorescence assay plate. AMCA fluorescence intensity was measured over time ($\lambda_{Ex}$ = 410 nm, $\lambda_{Em}$ = 490 nm). The 0.45 μM valinomycin was added to initiate the flux and fluorescence data were collected at 20-second intervals for 15 min, and then 1 μM CCCP was used to collapse the proton gradient. Data were processed according to established protocol (Su et al, 2016).

## Cell death analysis

Cells were treated to activate STING as described above. When obvious cell death was observed, cells were imaged by phase contrast microscopy followed by fixation and permeabilization. Cells were then stained with DAPI and imaged to collect the nuclear images for cell number quantification by a custom code. Dead or dying cells with irregular nuclear shapes were excluded automatically by the software. The total live cells were quantified in each condition and compared with the control cells without STING-dependent cell death.

## Image analysis

For the quantification of fluorescence images, the outlines of randomly selected cells were manually annotated in each image by an investigator blinded to the allocation. The fraction of protein A's intensity on protein B was quantified by comparing the sum of signal intensities of protein A overlapping with protein B to the total intensity of protein A in a given cell. When quantifying the fraction of protein A positive for protein B puncta, a higher threshold was applied to minimize diffuse protein B signals. Thresholds for both proteins were determined by the target protein's signal intensity percentile within a single cell, adjusted by a small constant. All cells from the same experiments were applied using the same threshold settings. Manual checking ensured accurate threshold applications. Quantitative data were exported to Excel and visualized using Prism to display mean and SEM values.

## Software

A Leica SP8 LIGHTNING confocal system and its built-in software Leica Application Suite X 3.5.5.19976 were used to collect confocal images as well as for living cell imaging. Images were processed in Adobe Photoshop 20.0.4. Schematic illustration figures were made with Biorender and Microsoft Office PowerPoint Professional Plus 2016. Protein structures were visualized in the PyMOL, Version 2.4.0. Graphs were generated in GraphPad Prism 9.0.0. Images were quantified using custom-coded algorithms tailored according to specific experiments to incorporate quality control measures, minimize errors, and ensure accuracy and reliability.

## Statistics and reproducibility

All experiments were independently reproduced at least three times unless otherwise indicated. Strict standards were applied to screen for robust and unbiased results. No statistical methods were used to predetermine sample size. The investigators were blinded to allocation during data collection and image quantification. Data were presented as mean ± SEM. Statistical significance was determined by unpaired, two-tailed $t$ tests.

## Data availability

Custom codes for image quantifications are available at https://github.com/jaytanlab/Protein_Colocalization_Quantification.

## Peer review information

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

## Acknowledgements

We thank members of the Shang and Tan laboratories and the Aging Institute members for discussions. We thank Drs. Michael B Butterworth and Daniel C Devor from University of Pittsburgh as well as Dr. Qingfeng Chen from Yunnan University for ion channel discussions. We thank Dr. Christian Rosenmund from Charité-Universitätsmedizin Berlin for the lyso-pHluorin plasmid. This work was supported by the National Key R&D Program of China (2021YFC2301400); the National Natural Science Foundation of China (32070876); the Shanxi Provincial Science Fund for Distinguished Young Scholars program (202103021221001) to GS. This publication was also supported by start-up funding from the Aging Institute at the University of Pittsburgh School of Medicine and the University of Pittsburgh Medical Center (UPMC) and a UPMC competitive medical research fund award to JXT. The revision of this work was partially supported by the National Institute of General Medical Sciences of the National Institutes of Health under award number R35GM150506 (JXT).

## Author contributions

**Jinrui Xun**: Data curation; Formal analysis; Investigation; Methodology; Writing—review and editing. **Zhichao Zhang**: Data curation; Formal analysis; Investigation; Methodology; Writing—review and editing. **Bo Lv**: Data curation; Formal analysis; Investigation; Methodology; Writing—review and editing. **Defen Lu**: Data curation; Formal analysis; Investigation; Methodology; Writing—review and editing. **Haoxiang Yang**: Data curation; Software; Formal analysis; Writing—review and editing. **Guijun Shang**: Conceptualization; Data curation; Formal analysis; Supervision; Funding acquisition; Investigation; Methodology; Writing—original draft; Project administration; Writing—review and editing. **Jay Xiaojun Tan**: Conceptualization; Data curation; Formal analysis; Supervision; Funding acquisition; Investigation; Methodology; Writing—original draft; Project administration; Writing—review and editing.

## Disclosure and competing interests statement

The authors declare no competing interests.

# Expanded View Figures

**Figure EV1.  STING traffics through the Golgi, causing morphological changes of the Golgi stacks.**

(A) Left, compound C53 inhibits diABZI-induced LC3 lipidation in BJ cells without blocking TBK1 signaling and STING phosphorylation. BJ cells treated as indicated were harvested 2 h after treatment for western blot. Right, quantification of the relative levels of LC3-II normalized to GAPDH in (A). Mean ± SEM; $n = 3$. (B) STING trafficking triggers morphological changes of the Golgi stacks 30 min after cGAMP stimulation in U2OS cells. Note brighter and swelled TGN marker Golgin97 at 30 min. Monoclonal U2OS Flag-hSTING cells were stimulated with 1 μM cGAMP and fixed at indicated time points for the co-staining of endogenous *cis*- and *trans*-Golgi markers GM130 and Golgin97, respectively. (C, D) Quantification of the colocalization between GM130 and Golgin97 in (B). Mean ± SEM; $n = 68$, 96, 50, and 86 random cells for 0′, 30′, 60′, and 90′, respectively. (E) STING trafficking triggers morphological changes of the Golgi stacks 30 min after cGAMP stimulation in HT1080 cells expressing endogenous STING. Note brighter and swelled TGN marker Golgin97 as well as increased overlap between GM130 and Golgin97 at 30 min. HT1080 cells were stimulated with 1 μM cGAMP and fixed at indicated time points for the co-staining of endogenous cis- and *trans*-Golgi markers GM130 and Golgin97, respectively. (F) STING trafficking causes TGN46 budding onto Golgi-derived vesicles which partially colocalize with post-Golgi STING vesicles. Monoclonal U2OS Flag-hSTING cells were stimulated with 1 μM cGAMP and fixed at indicated time points for the co-staining of STING and TGN46. Note, TGN46 was found out of the Golgi stacks at 30 min before STING budded out from TGN. (G) diABZI stimulates STING trafficking through the Golgi body. Monoclonal U2OS Flag-hSTING cells were stimulated with 1 μM diABZI and fixed at indicated time points for the co-staining of STING and Golgin97. (H) Quantification of the colocalization between STING and Golgin97 in (G). Mean ± SEM; $n = 57$, 65, 76, 74, and 66 random cells for 0′, 30′, 60′, 120′, and 180′, respectively. Data Information: Bar, 10 μm for all cell imaging panels. Statistical significance was determined by unpaired, two-tailed t tests for all quantifications. Source data are available online for this figure.

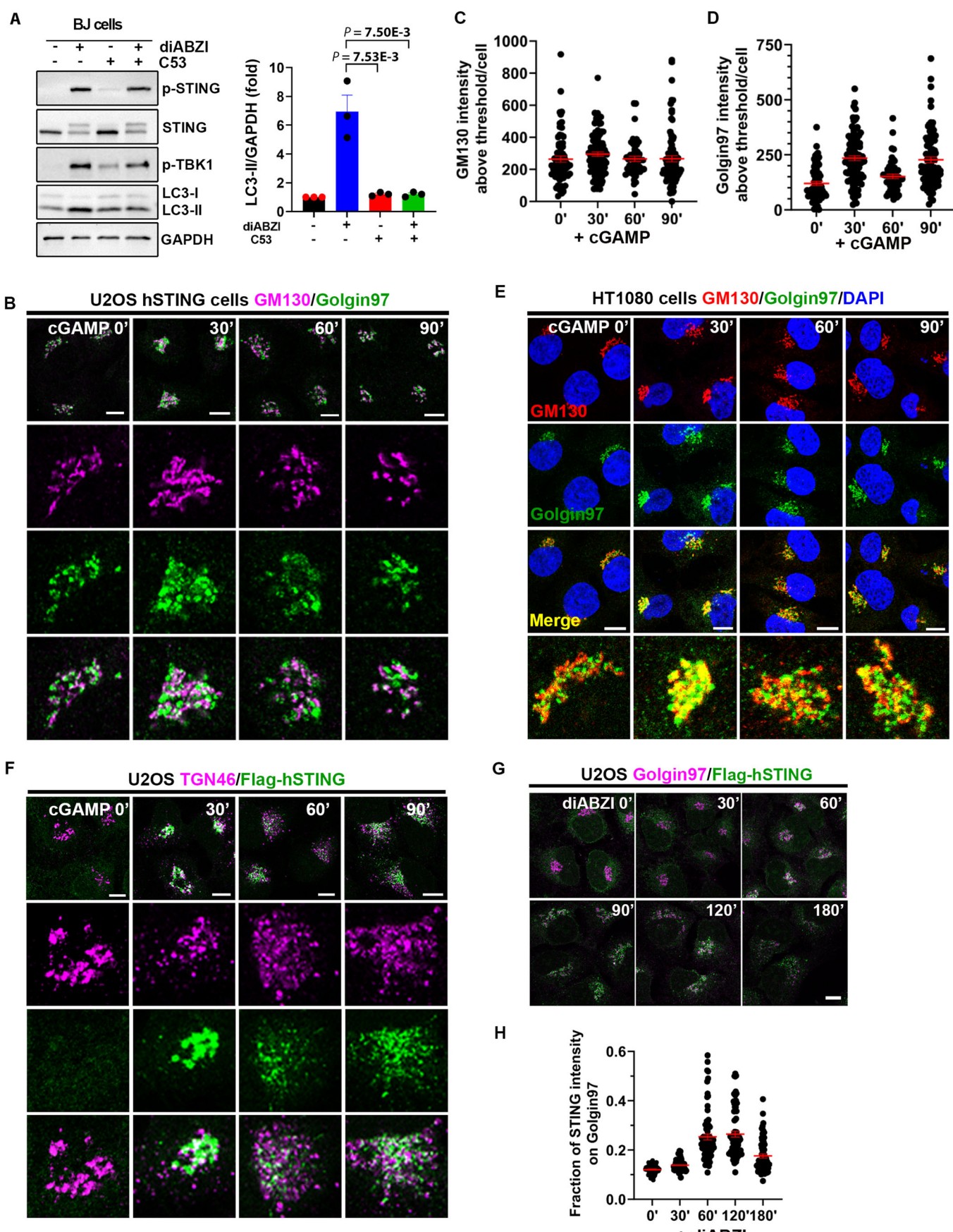

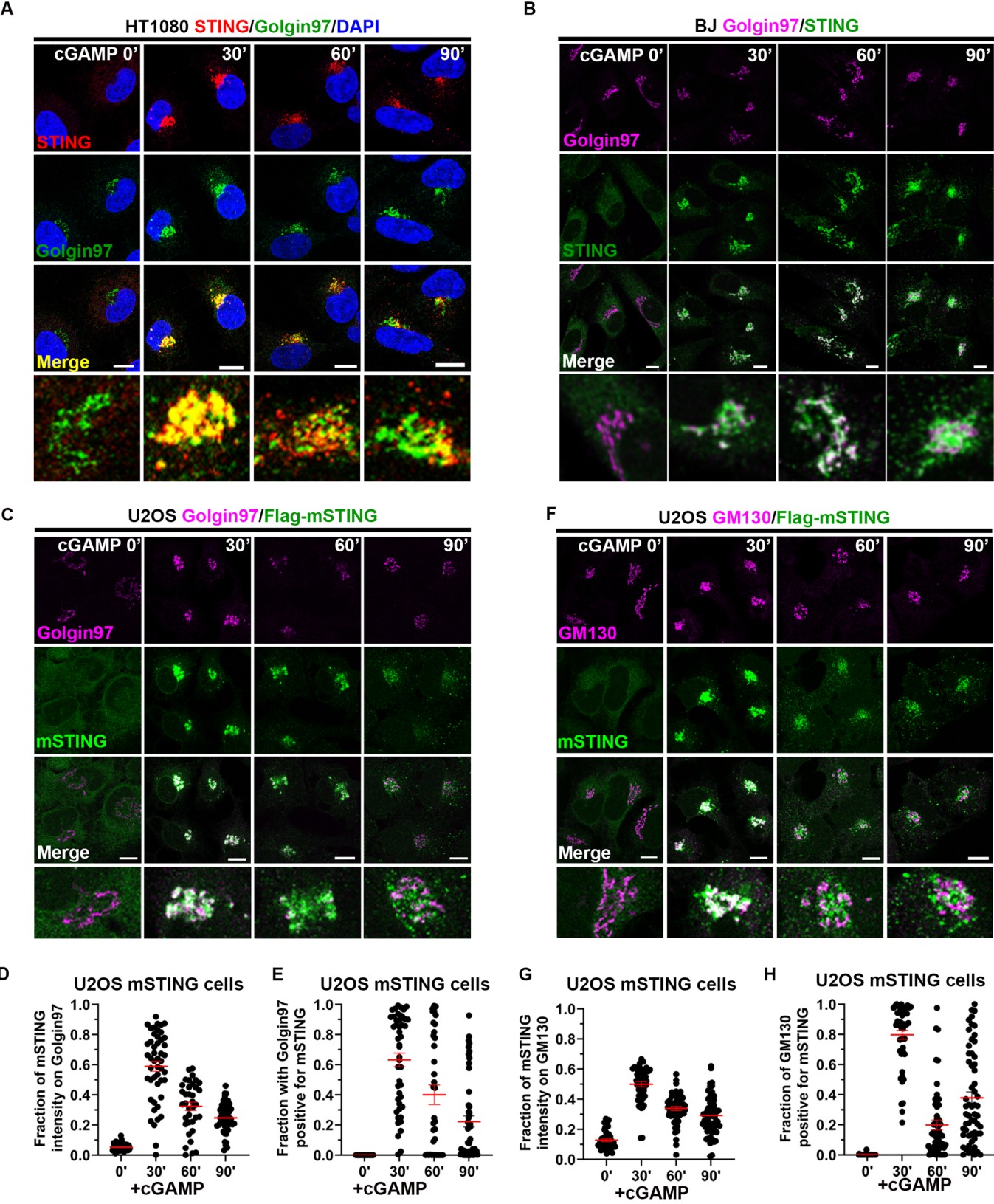

**◄**  **Figure EV2. Human and mouse STING similarly traffic through the Golgi complex.**

(A, B) STING traffics through the Golgi after cGAMP stimulation in HT1080 (A) and BJ (B) cells, both expressing endogenous human STING. Cells were stimulated with 1 μM cGAMP and fixed at indicated time points for the co-staining of endogenous STING and the *trans*-Golgi marker Golgin97, respectively. Note, brighter and swelled TGN marker Golgin97 at 30 min in HT1080 cells, whereas STING accumulation at the Golgi in BJ cells appeared to peak between 30 and 60 min. (C) Mouse STING traffics through the *trans*-Golgi upon cGAMP binding. U2OS cells stably expressing low levels of Flag-mSTING were stimulated with 1 μM cGAMP and fixed at indicated time points for the co-staining of STING and the *trans*-Golgi marker Golgin97. (D, E) Quantification of the colocalization between STING and Golgin97 in (C). Mean ± SEM; $n = 49, 48, 36$, and 52 random cells for 0′, 30′, 60′, and 90′, respectively. (F) STING traffics through the *cis*-Golgi upon cGAMP binding. U2OS cells stably expressing low levels of Flag-mSTING were stimulated with 1 μM cGAMP and fixed at indicated time points for the co-staining of STING and the cis-Golgi marker GM130. (G, H) Quantification of the colocalization between STING and GM130 in (F). Mean ± SEM; $n = 41, 50, 51$, and 59 random cells for 0′, 30′, 60′, and 90′, respectively. Data Information: Bar, 10 μm for all cell imaging panels. Source data are available online for this figure.

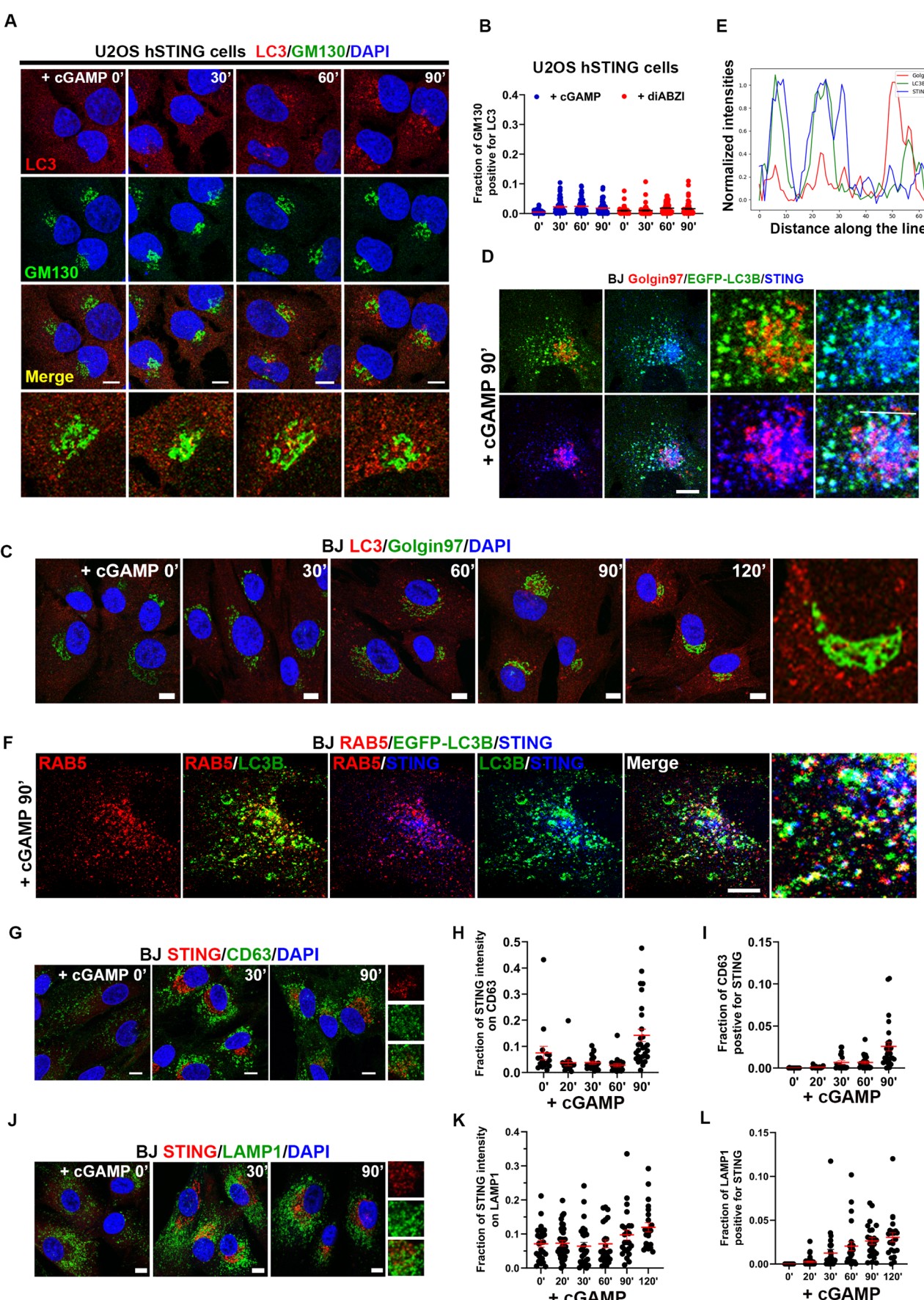

**◄**

**Figure EV3. Post-Golgi STING vesicles develop endosome-like properties accompanied by LC3 lipidation.**

(A) STING induces LC3 puncta outside of the Golgi. Monoclonal U2OS Flag-hSTING cells were stimulated with 1 μM cGAMP and fixed at indicated time points for the co-staining of endogenous LC3 and the *cis*-Golgi marker GM130. (B) Quantification of the colocalization between LC3 and GM130. Mean ± SEM; $n = 73, 79, 75, 86, 40, 55, 87$, and 71 random cells from left to right. (C) STING induces LC3 puncta outside of the Golgi in BJ cells. Cells were stimulated with 1 μM cGAMP and fixed at indicated time points for the co-staining of endogenous LC3 and the *trans*-Golgi marker Golgin97. (D) LC3 puncta were found on STING vesicles leaving the perinuclear vesicle clusters and were not found on Golgin97. BJ cells stably expressing EGFP-LC3B were fixed 90 min after cGAMP stimulation for immunostaining of STING and Golgin97. (E) EGFP-LC3B colocalizes with STING but not Golgin97. Normalized fluorescence intensities of Golgin97, EGFP-LC3B, and STING along the white line in the right bottom image of (D). (F) LC3 puncta were found on STING vesicles positive for RAB5, near the periphery of the post-Golgi vesicle clusters. BJ cells stably expressing EGFP-LC3B were fixed 90 min after cGAMP stimulation for immunostaining of STING and RAB5. (G) STING puncta develop a relatively low level of colocalization with the late endosome/lysosome marker CD63. BJ Cells were stimulated with 1 μM cGAMP and fixed at indicated time points for the co-staining of endogenous STING and CD63. (H, I) Quantification of the colocalization between STING and CD63 in (G). Mean ± SEM; $n = 16, 16, 20, 23$, and 28 random cells from left to right. (J) STING puncta develop a relatively low level of colocalization with the late endosome/lysosome marker LAMP1. BJ Cells were stimulated with 1 μM cGAMP and fixed at indicated time points for the co-staining of endogenous STING and CD63. (K, L) Quantification of the colocalization between STING and LAMP1 in (J). Mean ± SEM; $n = 27, 23, 37, 29, 24$, and 24 random cells from left to right. Data Information: Bar, 10 μm for all cell imaging panels. Source data are available online for this figure.

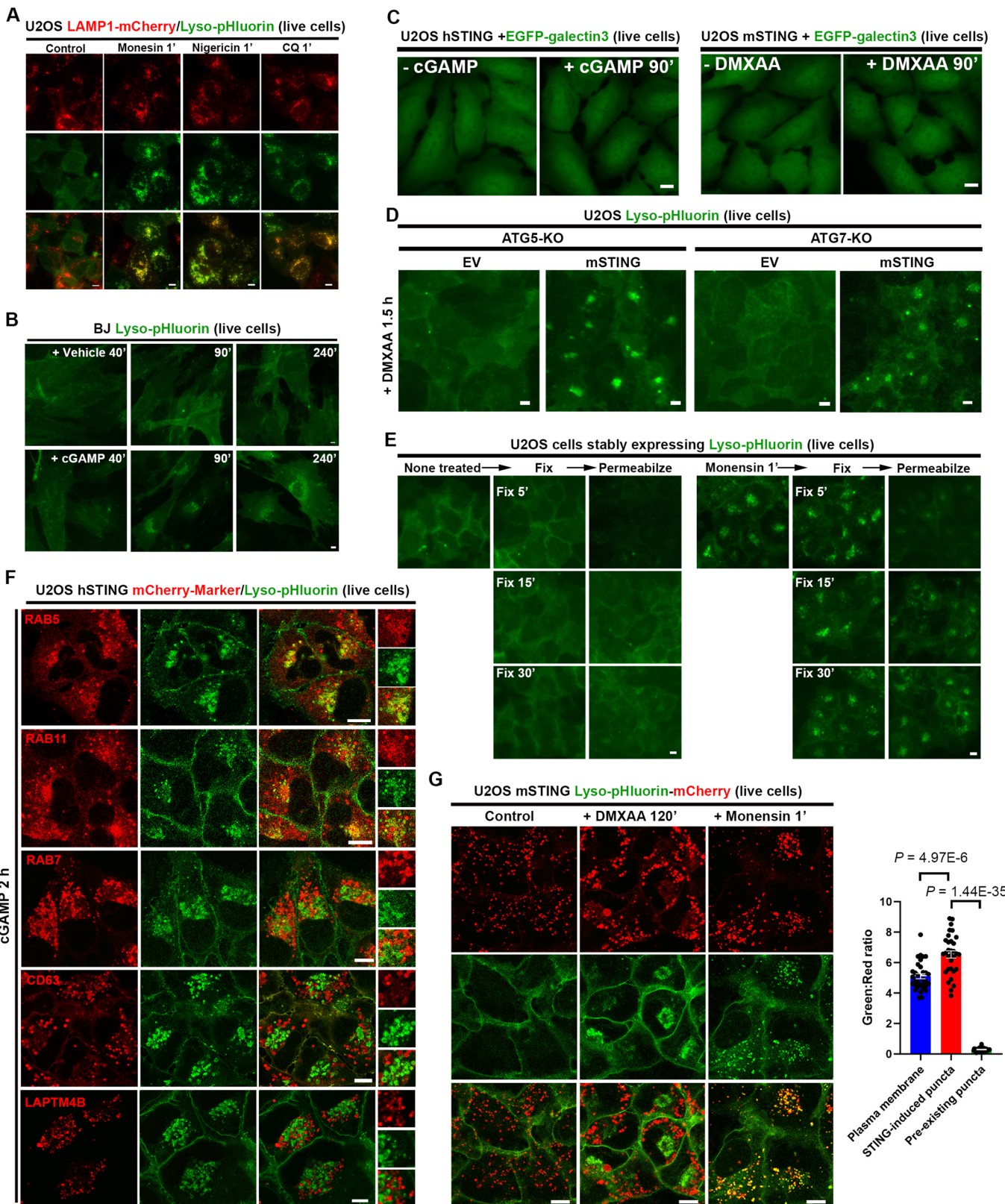

◀ **Figure EV4. STING neutralizes the pH of post-Golgi vesicles, which is captured by lyso-pHluorin, an endolysosomal pH sensor.**

(A) Validation of Lyso-pHluorin as a pH probe by proton ionophores (monensin and nigericin) and weak base chloroquine (CQ). U2OS cells stably expressing lyso-pHluorin and LAMP1-mCherry were treated with monensin, nigericin, or CQ for 1 min, and the Lyso-pHluorin puncta were monitored by wide-field live-cell imaging. (B) cGAMP stimulates lyso-pHluorin puncta in wild-type BJ cells. BJ cells stably expressing lyso-pHluorin were treated with digitonin buffer alone (Vehicle) or with cGAMP for 10 min, changed back to original media, and chased for indicated time periods. Lyso-pHluorin puncta were monitored by wide-field live-cell imaging. (C) STING activation does not induce EGFP-galectin3 puncta. U2OS cells stably expressing EGFP-galectin3 and human or mouse STING (hSTING/mSTING) were stimulated with indicated STING agonists. The fluorescence of EGFP-galectin3 were monitored by wide-field live-cell imaging. (D) Normal Lyso-pHluorin puncta formation upon mSTING activation by DMXAA in ATG5-KO (left) and ATG7-KO (right) U2OS cells stably expressing mSTING and lyso-pHluorin. Images taken using wide-field live-cell imaging. (E) Setting up a protocol to fix lyso-pHluorin cells for the immunostaining of other proteins without triggering new lyso-pHluorin puncta by fixation or permeabilization. Cells were seeded three days before treatment. Monensin was used to induce pre-existing puncta before fixation. 30 min of fixation in 4% electron microscopy grade polyformaldehyde (PFA) followed by 2 min of permeabilization in 0.1% Triton X-100 in PBS were used as a standard protocol to examine the colocalization of lyso-pHluorin puncta with other proteins by immunostaining. Images in this panel were taken using wide-field microscopy. (F) Live-cell images showing STING-induced lyso-pHluorin puncta together with mCherry-tagged organelle markers. U2OS cells stably expressing hSTING, lyso-pHluorin, and mCherry-tagged organelle markers were stimulated with cGAMP and subjected to confocal live-cell imaging. LAPTM4B, lysosomal-associated protein transmembrane 4 beta. (G) Detecting STING-induced vesicle deacidification by Lyso-pHluorin-mCherry. Left, U2OS cells stably expressing lyso-pHluorin-mCherry were stimulated as indicated and subjected to confocal live-cell imaging. Right, quantification of the green: red signal ratios at different subcellular localizations. Mean ± SEM; $n = 32$ random measurements for each subcellular localization. Data Information: Bar, 10 μm for all cell imaging panels. Statistical significance was determined by unpaired, two-tailed t tests for all quantifications. Source data are available online for this figure.

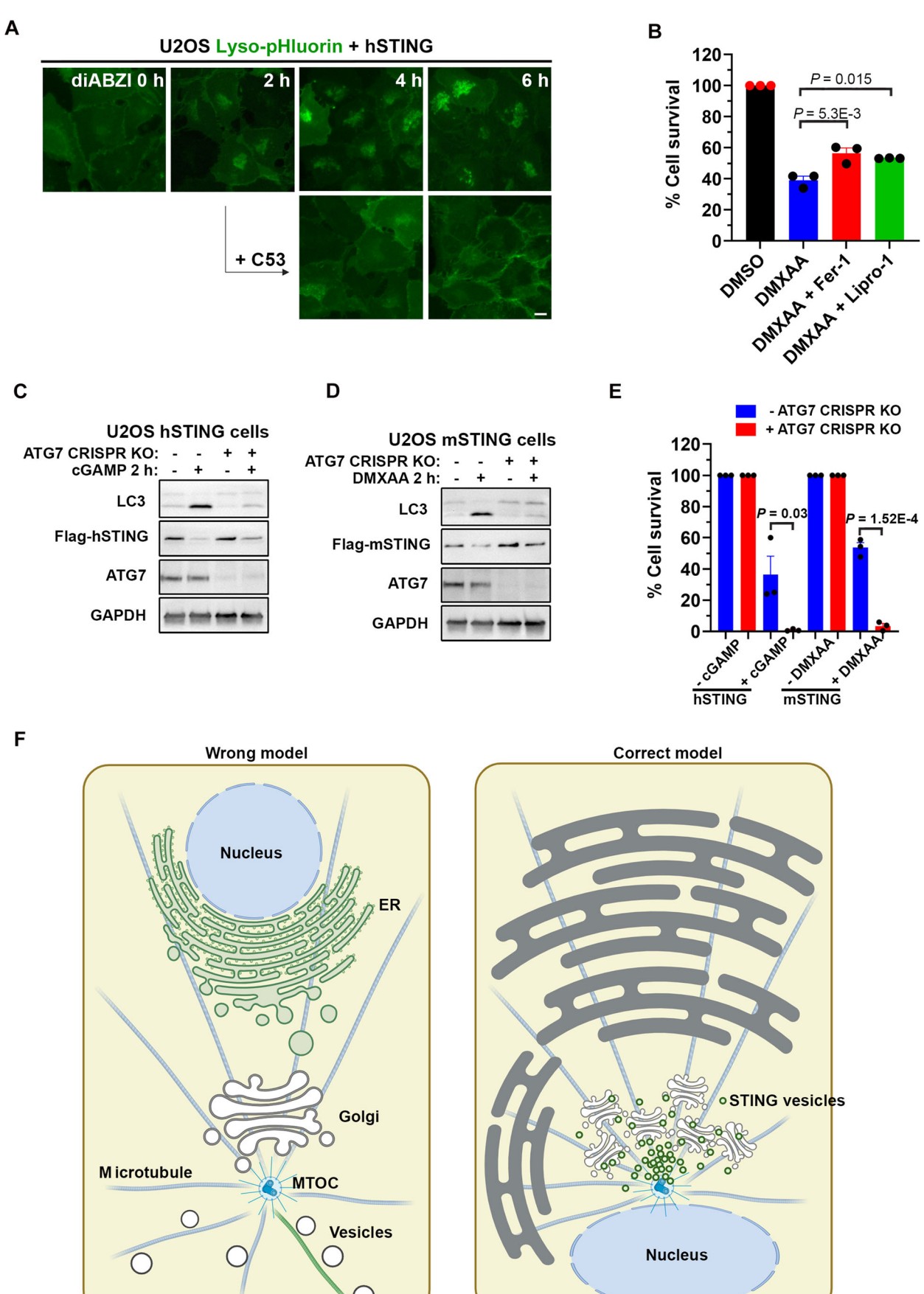

**A** U2OS Lyso-pHluorin + hSTING

diABZI 0 h    2 h    4 h    6 h

+ C53

**B**

% Cell survival

P = 0.015

P = 5.3E-3

DMSO    DMXAA    DMXAA + Fer-1    DMXAA + Lipro-1

**C** U2OS hSTING cells

ATG7 CRISPR KO: - - + +
cGAMP 2 h: - + - +

LC3
Flag-hSTING
ATG7
GAPDH

**D** U2OS mSTING cells

ATG7 CRISPR KO: - - + +
DMXAA 2 h: - + - +

LC3
Flag-mSTING
ATG7
GAPDH

**E**

% Cell survival

- ATG7 CRISPR KO
+ ATG7 CRISPR KO

P = 0.03

P = 1.52E-4

- cGAMP  + cGAMP  - DMXAA  + DMXAA
hSTING            mSTING

**F**

Wrong model

Nucleus
ER
Golgi
Microtubule    MTOC
Vesicles

Correct model

STING vesicles
Nucleus

**Figure EV5. Characterizing noncanonical functions of STING mediated through its transmembrane ion channel.**

(A) C53 addition 2 h after diABZI treatment strongly suppresses and reverses STING-induced lyso-pHluorin puncta. U2OS cells stably expressing hSTING and lyso-pHluorin were stimulated as indicated and the lyso-pHluorin puncta were monitored by live-cell imaging. Quantified in Fig. 7C. (B) Ferroptosis inhibitors ferrostatin-1 (Fer-1), liprostatin-1 (Lipro-1) partially block STING-dependent cell death. U2OS cells stably expressing mSTING were treated as indicated and cell death was analyzed 24 h after treatment. Mean ± SEM; $n = 3$. (C, D) ATG7-KO blocks STING-induced LC3 lipidation. U2OS cells stably expressing human (C) or mouse (D) STING were treated with or without ATG7 CRISPR KO lentiviruses for more than 7 days, and the KO pools were further treated as indicated to activate STING. Cells were harvested 2 h after treatment for western blot analysis of LC3 lipidation. (E) ATG7-KO accelerates STING-dependent cell death. The same cells from (C, D) were treated as indicated and cell death was analyzed 24 h after treatment. Mean ± SEM; $n = 3$. (F) Schematic illustration of the relative localization of the key organelles involved in STING trafficking, from the ER to the Golgi to post-Golgi endosome-like vesicles, as well as microtubules that may facilitate STING trafficking. Left, a typical wrong online carton model of the same set of organelles, which limits ER to the perinuclear region. Right, the correct model showing ER extension throughout the cytoplasm with the Golgi complex assembled as a cluster of Golgi stacks around the microtubule-organization center (MTOC). Note that Golgi itself is also considered an MTOC. Activated STING traffics through the Golgi to form post-Golgi vesicle clusters, after which STING gradually traffics out of this cluster for lysosomal degradation. The post-Golgi STING vesicles are packed close to the Golgi complex and caution is needed when interpreting the localization of these perinuclear STING vesicles. Data Information: Bar, 10 μm for all cell imaging panels. Statistical significance was determined by unpaired, two-tailed t tests for all quantifications. Source data are available online for this figure.

