## [Peer Review File · EMBO Reports]

A conserved ion channel function of STING mediates non-canonical autophagy and cell death

Jinrui Xun, Zhichao Zhang, Bo Lv, Defen Lu, Haoxiang Yang, Guijun Shang, and Jay Xiaojun Tan

DOI: 10.15252/embr.202358225

Corresponding author(s): Jay Xiaojun Tan (jay.tan@pitt.edu), Guijun Shang (guijun_shang@saari.org.cn)

Review Timeline:

Transfer Date:	26th Sep 23
Editorial Decision:	27th Sep 23
Revision Received:	23rd Oct 23
Editorial Decision:	6th Dec 23
Revision Received:	12th Dec 23
Accepted:	14th Dec 23

Transaction Report: This manuscript was transferred to EMBO reports following peer review at The EMBO Journal.

Referee #1:

In the paper by Xun et al, authors report an ion channel activity for STING, which is required for its activation of non-canonical autophagy and cell death. STING activation is known to initiate innate immune signalling through TBK1, and had previously been reported to induce non-canonical LC3 lipidation; although the mechanism of activation of LC3 lipidation was not known. Here authors report that following STING activation and trafficking from ER to the golgi, an ion channel property of STING is induced in post golgi - endosome like vesicles that results in their deacidification. This neutralisation of pH triggers non-canonical autophagy via the ATG16L1-V-ATPase axis.

In preparation of this manuscript, many of the key findings were published by another group (PMID: 37535724), including the ion channel property of STING, ion channel dependent deacidification of STING vesicles and subsequent activation of non-canonical autophagy. Blockade of STING ion channel activity with C53 was shown to inhibit activation of non-canonical autophagy.

In this manuscript, authors highlight further insights into the post-golgi trafficking of STING and how this impacts activation of LC3 lipidation. Specifically, they distinguish between the effect of C53 in blocking STING trafficking versus ion channel activity.

This is a well-executed set of experiments with clear conclusions. There is clearly value in solidifying these interesting findings regarding novel activities of STING. However, there are obviously questions on novelty.

Comments

1. Authors include further work on STING trafficking to distinguish their work from that previously published (PMID: 37535724). However, there is much work already published showing the transit of STING from ER, through the golgi and to the endolysosomal system, much of which is cited in this manuscript. So, it is less clear as to what extra this brings. The only extra information is that LC3 lipidation occurs on post golgi vesicles rather than the golgi itself.

2. Can authors block the movement of STING into post-golgi vesicles and, this increasing the dwell time at the golgi, and see if LC3 lipidation now occurs at the golgi membranes? Or is the lipidation really specific for the post golgi compartment. This was previously achieved by blocking GCC2 and different Rab GTPases (PMID: 36379959).

3. Can the authors provide any evidence that STING ion channel activation of LC3 lipidation on endolysosomal vesicles is in anyway related to the degradation of STING following activation?

4. The final section on ion channel activity dependent effects on cell death seem not to be linked with the previous work on LC3 lipidation. Is STING induced cell death affected by blocking LC3 lipidation?

5. The pH sensor used (Lyso-pHluorin), targets to late endosome/lysosomes via CD63. This would likely be separate from the Rab5 compartment where authors mainly locate STING. Is there any evidence that the robust Lyso-pHluorin signal overlaps with Rab5?

6. On Line 485 there is a mis-spelling of lipidation.

Referee #2:

In the manuscript, the authors demonstrated that agonist-bound STING can act as a proton channel and mediates non-canonical lipidation of LC-3. They claimed that this function of STING is a primary cause of cell death, which is often accompanied with STING activation in several cell types. The biochemical part in the study was sound, and this reviewer thinks it is worth reporting despite the same content has recently been published in Science. In contrast, cell biological part was underdeveloped, and requires thorough experiments to make the conclusion clear.

(1) They used Lyso-pHluorin throughout the study. I see several problems with this imaging.

* The images with Lyso-pHluorin are not consistent. For example, Fig. 4C (mSTING, 0 h) vs Fig. 4G (without stimulation). The former showed even distribution in cells, the latter showed the strong staining at periphery.

* This probe does not have the internal fluorescence control, making it hard to exclude the possibility that the increase in the intensity may simply reflect the amount of the probe. The experiments should be performed with ratio-type pH probe. With the ratio-type pH probe, the actual pH value can also be measured accurately.

* The probe should be validated anyway with bafilomycin A or chloroquine. The positive control is missing.

* The subcellular compartment in which the pH increases should be determined live.

(2) The recent study showed that the post-Golgi traffic of STING to lysosomes involves recycling endosomes, rather than early endosomes [Nat. Cell Biol. 25, 453 (2023)]. The authors should test Rab11 and/or transferrin receptor in Figure 3 and determine the site of LC3 lipidation "together with STING".

(3) The data in Fig. S1A, Fig. 6G should be quantitated.

(4) The effect of C53 addition to LC3 membrane localization should be provided (related to Fig 3D).

(5) The cell death data is interesting but needs more validation. What type of cell death is occurring? In Fig. S5, they found one mutant that accelerates the increase of pH. Does this mutant increase the rate of cell death?

Dear Jay,

Thank you for transferring your manuscript from The EMBO Journal to EMBO Reports. As discussed, I would like to invite you to revise your manuscript along the lines outlined in your revision plan and as discussed. We had penciled in a revision within 1 month, given the competitive situation.

Please also address all referee concerns in a complete point-by-point response. Acceptance of the manuscript will depend on a positive outcome of a second round of review, which I will fast-track.

2) individual production quality figure files as .eps, .tif, .jpg (one file per figure). Please download our Figure Preparation Guidelines (figure preparation pdf) from our Author Guidelines pages <https://www.embopress.org/page/journal/14693178/authorguide> for more info on how to prepare your figures.

4) a complete author checklist, which you can download from our author guidelines (<<https://www.embopress.org/page/journal/14693178/authorguide>>). Please insert information in the checklist that is also reflected in the manuscript. The completed author checklist will also be part of the RPF.

5) Please note that all corresponding authors are required to supply an ORCID ID for their name upon submission of a revised manuscript (<<https://orcid.org/>>). Please find instructions on how to link your ORCID ID to your account in our manuscript tracking system in our Author guidelines (<<https://www.embopress.org/page/journal/14693178/authorguide#authorshipguidelines>>)

6) We replaced Supplementary Information with Expanded View (EV) Figures and Tables that are collapsible/expandable online. A maximum of 5 EV Figures can be typeset. EV Figures should be cited as 'Figure EV1, Figure EV2' etc... in the text and their respective legends should be included in the main text after the legends of regular figures.

7) Please note that a Data Availability section at the end of Materials and Methods is now mandatory. In case you have no data that requires deposition in a public database, please state so instead of refereeing to the database. See also < <https://www.embopress.org/page/journal/14693178/authorguide#dataavailability>>. Please note that the Data Availability Section is restricted to new primary data that are part of this study.

Additional information on source data and instruction on how to label the files are available <<https://www.embopress.org/page/journal/14693178/authorguide#sourcedata>>.

10) Figure legends and data quantification:
The following points must be specified in each figure legend:

- the name of the statistical test used to generate error bars and P values,
 - the number (n) of independent experiments (please specify technical or biological replicates) underlying each data point,
 - the nature of the bars and error bars (s.d., s.e.m.)
-
- If the data are obtained from n {less than or equal to} 5, show the individual data points in addition to the SD or SEM.
 - If the data are obtained from n {less than or equal to} 2, use scatter blots showing the individual data points.

11) Our journal encourages inclusion of *data citations in the reference list* to directly cite datasets that were re-used and obtained from public databases. Data citations in the article text are distinct from normal bibliographical citations and should directly link to the database records from which the data can be accessed. In the main text, data citations are formatted as follows: "Data ref: Smith et al, 2001" or "Data ref: NCBI Sequence Read Archive PRJNA342805, 2017". In the Reference list, data citations must be labeled with "[DATASET]". A data reference must provide the database name, accession number/identifiers and a resolvable link to the landing page from which the data can be accessed at the end of the reference. Further instructions are available at <<https://www.embopress.org/page/journal/14693178/authorguide#referencesformat>>.

12) As part of the EMBO publication's Transparent Editorial Process, EMBO Reports publishes online a Review Process File to accompany accepted manuscripts. This File will be published in conjunction with your paper and will include the referee reports, your point-by-point response and all pertinent correspondence relating to the manuscript.

Kind regards,

Martina

We thank the reviewers and the editor for their supportive and insightful comments which have further improved the quality of this study. The additional experiments performed during the revision further strengthened our conclusions that STING de-acidifies endosome-like vesicles for LC3 lipidation, a new function that depends on the channel function of STING which also triggers cell death.

Here is a quick summary of the new data added to the revised manuscript:

- (1) Confirmed that STING-induced Lyso-pHluorin puncta are positive for RAB5 (**Fig 4K**). Although Lyso-pHluorin is based on a late endosome marker CD63, the fraction of the probe de-quenched by STING is not on pre-existing late endosome/lysosomes (**Fig EV4F, G**).
- (2) The Lyso-pHluorin probe, which was previously validated by others (Rost, Schneider et al., 2015b) and our group (Tan & Finkel, 2022), has now been further validated in the current manuscript (**Fig EV4A, E, G**).
- (3) We have further characterized Lyso-pHluorin puncta formation in live cell imaging, which confirmed that the vesicles de-acidified by STING are endosome-like compartments, with more colocalization with early and recycling endosome markers RAB5 and RAB11, and less with late endosome/lysosome markers RAB7, CD63, and LAPTM4B (**Fig EV4F**, quantified in **Fig 4L**).
- (4) More experiments testing STING-dependent cell death: the hyperactive channel mutant of STING (L54E) triggers more cell death than wild-type STING (**Fig 7F, G**)

See below for our point-by-point responses to all comments.

Editor's comments:

Please address all referee concerns and provide all controls required. It will be important to verify the compartment in which STING resides (early vs late endosomes) and whether LC3 lipidation occurs specifically in post-Golgi vesicles, to verify the Lyso-pHluorin probe and to strengthen the functional data on cell death.

1. Verify the compartment in which STING resides (early vs late endosomes)

STING colocalization with the early endosome marker RAB5 and late endosome/lysosome markers CD63/LAMP1 were quantified in our original manuscript (currently **Fig 3F/G/H, EV3F-L**), which indicated more colocalization with RAB5. We have now also quantified the colocalization of STING with the recycling endosome marker RAB11 (**Fig 4I/J/K**), as suggested by reviewer 2. Consistent with the recent work from Tomohiko Taguchi's group (Kuchitsu, Mukai et al., 2023), we observed a strong colocalization between STING and RAB11. We further quantified the colocalization of STING-induced Lyso-pHluorin puncta with different endosome/lysosome markers (**Fig EV4F**, quantified in **Fig 4L**). This confirmed that these Lyso-pHluorin puncta mostly overlapped with both the early and recycling endosome markers. We concluded that STING vesicles that are de-acidified represent endosome-like post-Golgi vesicles.

2. Whether LC3 lipidation occurs specifically in post-Golgi vesicles

Our data indicate that the endogenous LC3 puncta induced by STING has little overlap with the Golgi complex. However, we noticed that, when EGFP-LC3B was overexpressed, a low level of EGFP-LC3B signal was detected on the Golgi 30 min after STING activation - when most STING was still on the Golgi (**Fig R1**). Thus, it seems the Golgi membrane is a minor site of LC3 lipidation downstream of STING, which can be detected when LC3 is overexpressed. It is possible that blocking STING at the Golgi can cause more LC3 lipidation on the Golgi, but this does not seem to change our conclusion that the endogenous LC3 puncta mostly occurred in post-Golgi vesicles.

3. Verify the Lyso-pHluorin probe

This probe was verified using a proton ionophore monensin in our original submission (currently **Fig EV4E**) as well as by other lysosomal pH neutralization methods in other studies (Rost, Schneider et al., 2015a, Tan & Finkel, 2022). It has now been further validated with more experiments in the revised manuscript (**Fig EV4A, E, G**).

4. Strengthen the functional data on cell death.

We have added more cell death tests as suggested by the reviewers. However, we believe the reviewers might agree that although cell death has been found to depend on the channel of STING, the exact mechanism by which the channel function kills the cell is beyond the scope of this study.

- a. New experiments using LC3-lipidation-deficient cells ruled out the requirement of LC3 lipidation in STING-dependent cell death. We used two different ways to block STING-dependent LC3 lipidation: (1) expression of a bacterial effector protein SopF (**Fig 7H/I/J**) which modifies V-ATPase to block ATG16L1 recruitment; (2) ATG7 depletion by CRISPR KO (**Fig EV7C/D/E**).
- b. The impact of the L54E mutation on STING-dependent cell death has been tested, which showed more cell death downstream of this mutant compared with the wild-type STING expressed in the same level (**Fig 7F/G**).
- c. We also tested whether ferroptosis (Wu, Liu et al., 2022) is a major mechanism for STING-dependent cell death using two different ferroptosis inhibitors. However, ferroptosis appeared to play a minor role (**Fig EV7B**).

Referee #1:

In the paper by Xun et al, authors report an ion channel activity for STING, which is required for its activation of non-canonical autophagy and cell death. STING activation is known to initiate innate immune signalling through TBK1, and had previously been reported to induce non-canonical LC3 lipidation; although the mechanism of activation of LC3 lipidation was not known. Here authors report that following STING activation and trafficking from ER to the golgi, an ion channel property of STING is induced in post golgi - endosome like vesicles that results in their deacidification. This neutralisation of pH triggers non-canonical autophagy via the ATG16L1-V-ATPase axis.

In preparation of this manuscript, many of the key findings were published by another group (PMID: 37535724), including the ion channel property of STING, ion channel dependent deacidification of STING vesicles and subsequent activation of non-canonical autophagy. Blockade of STING ion channel activity with C53 was shown to inhibit activation of non-canonical autophagy.

In this manuscript, authors highlight further insights into the post-golgi trafficking of STING and how this impacts activation of LC3 lipidation. Specifically, they distinguish between the effects of C53 in blocking STING trafficking versus ion channel activity.

This is a well-executed set of experiments with clear conclusions. There is clearly value in solidifying these interesting findings regarding novel activities of STING. However, there are obviously questions on novelty.

We thank this reviewer for the supportive comments.

Comments

1. Authors include further work on STING trafficking to distinguish their work from that previously published (PMID: 37535724). However, there is much work already published showing the transit of STING from ER, through the golgi and to the endolysosomal system, much of which is cited in this manuscript. So, it is less clear as to what extra this brings. The only extra information is that LC3 lipidation occurs on post golgi vesicles rather than the golgi itself.

We thank the reviewer for bringing this important point. We aimed to clarify in Fig. 2 that STING traffics through the Golgi to form post-Golgi vesicle clusters instead of staying in the Golgi body for signaling purpose. We have now added discussions about this to the revised manuscript.

The literature is controversial regarding the signaling compartments of STING (ERGIC, Golgi, or post-Golgi vesicles). Several reasons might have contributed to such controversy: **(1)** the post-Golgi STING vesicles form clusters around the microtubule organization center (MTOC) where the Golgi body is also located and considered as part of the MTOC (**Fig. R2, right; Fig EV7F** in the revised manuscript). The partial overlap between the Golgi and the post-Golgi STING vesicle clusters likely led to some mis-interpretation that STING stays on the Golgi for signaling purpose; **(2)** Some trans-Golgi markers traffic out of the Golgi (e.g., TGN46) upon STING activation, which makes the tracking dynamics of STING more complicated; **(3)** lower resolution confocal images precluded the differentiation between the Golgi and post-Golgi vesicle clusters; **(4)** STING first traffics through the Golgi to the perinuclear vesicle clusters, after which it again slowly traffics out of this region for lysosomal degradation (**Fig R2, right**). Both steps happen around the MTOC area, which could cause misunderstanding of this process. **(5)** Wrong online cartoon models of the relative localizations of key subcellular organelles (**Fig. R2, left**).

Fig R2. Schematic illustration of STING trafficking to the area around the perinuclear microtubule organization center (MTOC). **Left:** wrong model of the relative localizations of key organelles involved in STING trafficking. **Right:** the correct model.

2. Can authors block the movement of STING into post-golgi vesicles and, this increasing the dwell time at the golgi, and see if LC3 lipidation now occurs at the golgi membranes? Or is the lipidation really specific for the post golgi compartment. This was previously achieved by blocking GCC2 and different Rab GTPases (PMID: 36379959).

We appreciate these discussions and suggestions. The LC3 lipidation down stream of STING did not seem to be specific to post-Golgi membranes, because we noticed that when we overexpressed EGFP-LC3B, the earliest, weak EGFP-LC3B puncta colocalized with the Golgi, but stronger puncta developed afterwards outside the Golgi (**Fig R1** on Page 1 of this document). It is thus likely that a relatively small fraction of LC3 lipidation happens on the Golgi. However, endogenous LC3 puncta never showed obvious colocalization with Golgi markers, no matter STING is endogenous or ectopic (**Fig EV3A, C**). It is possible that upon STING retention on the Golgi, LC3 lipidation might now occur at the Golgi membrane. However, this does not seem to change our conclusion that most LC3 puncta were detected on post-Golgi, endosome-like STING vesicles. We have now added discussions about this in the revised manuscript.

3. Can the authors provide any evidence that STING ion channel activation of LC3 lipidation on endolysosomal vesicles is in anyway related to the degradation of STING following activation?

We have previously shown that ATG5-KO cells do not have major defects in STING degradation (Gui, Yang et al., 2019). Thus, we do not expect a major impact of LC3 lipidation on STING degradation. Instead, recent data from multiple other groups have shown that STING vesicle clusters are directly delivered to lysosomes through endosomal sorting complex required for transport (ESCRT)-mediated micro-autophagy (Balka, Venkatraman et al., 2023, Gentili, Liu et al., 2023, Kuchitsu et al., 2023).

4. The final section on ion channel activity dependent effects on cell death seem not to be linked with the previous work on LC3 lipidation. Is STING induced cell death affected by blocking LC3 lipidation?

STING has several non-canonical functions including LC3 lipidation and cell death. We asked whether these functions relied on the new channel function of STING. Our data showed that both LC3 lipidation and cell death were fully dependent on the channel of STING. However, LC3 lipidation is not required for cell death. We tested two different methods to block STING-mediated LC3 lipidation, neither of which suppressed STING-dependent cell death.

(1) Overexpression of a bacterial effector protein SopF, which is known to block V-ATPase-mediated ATG16L1 recruitment downstream of STING (Fischer, Wang et al., 2020), strongly suppressed STING-mediated LC3 lipidation (**Fig 7H/I**). However, SopF failed to block cell death triggered by either human or mouse STING (**Fig 7J**).

(2) Similar to SopF overexpression, ATG7 CRISPR KO showed a robust block of STING-dependent LC3 lipidation (**Fig EV7C/D**). However, these KO cells consistently showed more cell death than control cells (**Fig EV7E**).

Thus, STING-dependent cell death is not mediated through LC3 lipidation, but both cell death and LC3 lipidation required the channel of STING.

5. The pH sensor used (Lyso-pHluorin), targets to late endosome/lysosomes via CD63. This would likely be separate from the Rab5 compartment where authors mainly locate STING. Is there any evidence that the robust Lyso-pHluorin signal overlaps with Rab5?

We also initially thought they were late endosome/lysosomes, but these Lyso-pHluorin puncta had little overlap with late endosome/lysosomes. The Lyso-pHluorin puncta colocalized well with both STING and RAB5 (**Fig 4K**, three channel colocalization). We have now cloned mCherry-tagged RAB5, RAB7, RAB11, CD63, and LAPTM4B, and compared their colocalization with STING-induced Lyso-pHluorin puncta through live cell imaging. Lyso-pHluorin puncta showed the highest overlap with RAB5 and RAB11, with less association with late

endosome/lysosome markers such as RAB7, CD63, and LAPTM4B (**Fig EV4F**, quantified in **Fig 4L**). Even when mCherry is directly fused to the C-terminal end of CD63 in the Lyso-pHluorin construct (Lyso-pHluorin-mCherry), the STING-induced pHluorin puncta still showed on vesicles separated from the pre-existing mCherry-positive late endosome/lysosomes (**Fig EV4G**). Thus, STING-induced pHluorin puncta come from a new pool of Lyso-pHluorin that is localized on STING vesicles which are different from the pre-existing late endosome/lysosomes. These STING vesicles developed endosome-like properties as they showed overlap with RAB5 and RAB11.

6. On Line 485 there is a mis-spelling of lipidation.

Corrected. Thanks!

Referee #2:

In the manuscript, the authors demonstrated that agonist-bound STING can act as a proton channel and mediates non-canonical lipidation of LC-3. They claimed that this function of STING is a primary cause of cell death, which is often accompanied with STING activation in several cell types. The biochemical part in the study was sound, and this reviewer thinks it is worth reporting despite the same content has recently been published in Science. In contrast, cell biological part was underdeveloped, and requires thorough experiments to make the conclusion clear.

(1) They used Lyso-pHluorin throughout the study. I see several problems with this imaging.

* The images with Lyso-pHluorin are not consistent. For example, Fig. 4C (mSTING, 0 h) vs Fig. 4G (without stimulation). The former showed even distribution in cells, the latter showed the strong staining at periphery.

The images in **Fig. 4G** were collected with confocal, whereas **Fig. 4A/C/E** were from wide-field, non-confocal microscope which detects more background signal from the plasma membrane. This has now been clarified in the figure legends.

* This probe does not have the internal fluorescence control, making it hard to exclude the possibility that the increase in the intensity may simply reflects the amount of the probe. The experiments should be performed with ratio-type pH probe. With the ratio-type pH probe, the actual pH value can also be measured accurately.

This is an important comment that we addressed through several points:

- (1) The observed Lyso-pHluorin puncta were not triggered by increased expression of the probe. The live cell imaging data in **Fig. 4G/H** showing a specific increase in the Lyso-pHluorin puncta instead of the whole signal in the cell provided a strict control that the new puncta were not due to increased expression levels of the probe during STING activation.
- (2) There is a possibility that the probe level on STING vesicles might slowly increase over time, but such increase should not be visible without pH neutralization of these vesicles. First, these puncta did not occur if the STING channel is blocked (**Fig 6E/F**). Second, after the formation of these Lyso-pHluorin puncta, they can be re-quenched upon blocking of the STING channel by C53 (**Fig 7A**). Thus, the increased Lyso-pHluorin puncta were not simply due to a change of the probe levels on STING vesicles.
- (3) We generated a potential ratio-type pH probe by fusing mCherry to the C-terminal end of CD63 in the Lyso-pHluorin construct (Lyso-pHluorin-mCherry). The pHluorin sequence is inserted in the small luminal loop of CD63. We performed live cell imaging to check the relative signals of pHluorin and mCherry before and after STING activation. STING-induced pHluorin puncta overlapped with very weak, new mCherry signal outside of the pre-existing mCherry vesicles on late endosome/lysosomes (**Fig EV4G**, left and middle). As a positive control for de-acidification-induced pHluorin puncta in this experiment, we detected robust pHluorin puncta that overlaps with pre-existing mCherry vesicles upon the addition of a proton ionophore monensin (**Fig EV4G**, right).

Thus, STING-induced Lyso-pHluorin puncta likely represent a fraction of the probe localized to STING vesicles. The fluorescence signal of this fraction of Lyso-pHluorin requires the channel of STING to de-acidify these vesicles.

* The probe should be validated anyway with bafilomycin A or chloroquine. The positive control is missing.

Yes, we have now fully validated the probe with chloroquine (CQ, membrane-permeable weak base), monensin (proton ionophore), and nigericin (proton ionophore). In our lab submission, we showed that Lyso-pHluorin puncta were immediately induced upon the addition of monensin (current **Fig EV4E**). We have now further added Lyso-pHluorin validation data with chloroquine and nigericin; lysosomal pH neutralization by different chemicals

consistently triggered robust Lyso-pHluorin puncta that colocalize with the lysosome marker LAMP1 (**Fig EV4A**). See also **Fig EV4G**.

* The subcellular compartment in which the pH increases should be determined live.

We have now cloned mCherry-tagged RAB5, RAB7, RAB11, CD63, and LAPTM4B, and compared their colocalization with STING-induced Lyso-pHluorin puncta through live cell imaging. Lyso-pHluorin puncta showed the highest overlap with RAB5 and RAB11, with less association with late endosome/lysosome markers such as RAB7, CD63, and LAPTM4B (**Fig EV4F**, quantified in **Fig 4L**). Even when mCherry is directly fused to the C-terminal end of CD63 in the Lyso-pHluorin construct (Lyso-pHluorin-mCherry), the STING-induced pHluorin puncta still showed on vesicles separated from the pre-existing mCherry-positive late endosome/lysosomes (**Fig EV4G**). Thus, STING-induced pHluorin puncta come from a different pool of Lyso-pHluorin that is localized on STING vesicles which are different from the pre-existing late endosome/lysosomes. These STING vesicles developed endosome-like properties as they showed overlap with RAB5 and RAB11. We believe these vesicles are not bona fide early or recycling endosomes, as they appear to be clusters of STING trafficking vesicles that recruit these RABs. Thus, we described them as endosome-like post-Golgi STING vesicles.

(2) The recent study showed that the post-Golgi traffic of STING to lysosomes involves recycling endosomes, rather than early endosomes [Nat. Cell Biol. 25, 453 (2023)]. The authors should test Rab11 and/or transferrin receptor in Figure 3 and determine the site of LC3 lipidation "together with STING".

Thanks for this important suggestion. We have examined STING/LC3 co-staining with EGFP-RAB11, which revealing extensive overlap between STING with RAB11, with a relatively less colocalization between LC3 and RAB11 (**Fig 3I/J/K**). As described above, we have also tried live cell imaging testing the colocalization between STING-induced Lyso-pHluorin puncta with mCherry-RAB11 as well as other organelle markers (**Fig EV4F**, quantified in **Fig 4L**). The Lyso-pHluorin puncta appeared to associate with both RAB5 and RAB11. STING vesicles might use these RAB proteins for their trafficking. Future studies are needed to fully define the properties of these vesicles, but we used "endosome-like vesicles" at this point to describe the vesicles de-acidified by STING.

(3) The data in Fig. S1A, Fig. 6G should be quantitated.

We have now added quantification of these panels.

(4) The effect of C53 addition to LC3 membrane localization should be provided (related to Fig 3D).

Yes, consistent with the western result, C53 fully blocks LC3 puncta formation in immunofluorescence. The new IF images have been added to **Fig 6I/J**.

(5) The cell death data is interesting but needs more validation. What type of cell death is occurring? In Fig. S5, they found one mutant that accelerates the increase of pH. Does this mutant increase the rate of cell death?

We have now compared the impact of STING-WT and -L54E on cell death. We expressed similar amounts of the two proteins in U2OS cells. L54E consistently triggered more cell death compared with STING-WT (**Fig 7F/G**), consistent with a role for proton release in STING-dependent cell death.

STING-dependent cell death likely involves multiple mechanisms including ferroptosis (Tang, Zundell et al., 2016, Wu, Chen et al., 2019, Wu et al., 2022, Xu, Chen et al., 2023). We found that two different ferroptosis inhibitors partially suppressed STING-dependent cell death (**Fig EV7B**). However, they were not able to fully block cell death, suggesting the presence of additional cell death mechanisms activated by STING. We further investigated whether this is a type of autophagic cell death as the channel also mediates non-canonical LC3 lipidation. Deletion of ATG7 or overexpression of a bacterial effector protein SopF strongly suppressed STING-dependent LC3 lipidation (**Fig 7H/I, EV7C/D**). However, neither of them blocked STING-dependent cell death (**Fig 7J, EV7E**). Thus, although the channel of STING mediates both LC3 lipidation and cell death, the two downstream processes appear to be independent of each other. We plan to further pursue the mechanistic studies regarding STING-dependent cell death as a future direction.

References:

- Balka KR, Venkatraman R, Saunders TL, Shoppee A, Pang ES, Magill Z, Homman-Ludiye J, Huang C, Lane RM, York HM, Tan P, Schittenhelm RB, Arumugam S, Kile BT, O'Keeffe M, De Nardo D (2023) Termination of STING responses is mediated via ESCRT-dependent degradation. *The EMBO Journal* 42: e112712
- Fischer TD, Wang C, Padman BS, Lazarou M, Youle RJ (2020) STING induces LC3B lipidation onto single-membrane vesicles via the V-ATPase and ATG16L1-WD40 domain. *Journal of Cell Biology* 219
- Gentili M, Liu B, Papanastasiou M, Dele-Oni D, Schwartz MA, Carlson RJ, Al'Khafaji AM, Krug K, Brown A, Doench JG, Carr SA, Hacohen N (2023) ESCRT-dependent STING degradation inhibits steady-state and cGAMP-induced signalling. *Nature Communications* 14: 611
- Gui X, Yang H, Li T, Tan X, Shi P, Li M, Du F, Chen ZJ (2019) Autophagy induction via STING trafficking is a primordial function of the cGAS pathway. *Nature* 567: 262-266
- Kuchitsu Y, Mukai K, Uematsu R, Takaada Y, Shinjima A, Shindo R, Shoji T, Hamano S, Ogawa E, Sato R, Miyake K, Kato A, Kawaguchi Y, Nishitani-Isa M, Izawa K, Nishikomori R, Yasumi T, Suzuki T, Dohmae N, Uemura T et al. (2023) STING signalling is terminated through ESCRT-dependent microautophagy of vesicles originating from recycling endosomes. *Nature Cell Biology* 25: 453-466
- Rost BR, Schneider F, Grauel MK, Wozny C, Bentz C, Blessing A, Rosenmund T, Jentsch TJ, Schmitz D, Hegemann P, Rosenmund C (2015a) Optogenetic acidification of synaptic vesicles and lysosomes. *Nat Neurosci* 18: 1845-1852
- Rost BR, Schneider F, Grauel MK, Wozny C, G Bentz C, Blessing A, Rosenmund T, Jentsch TJ, Schmitz D, Hegemann P, Rosenmund C (2015b) Optogenetic acidification of synaptic vesicles and lysosomes. *Nature Neuroscience* 18: 1845-1852
- Tan JX, Finkel T (2022) A phosphoinositide signalling pathway mediates rapid lysosomal repair. *Nature* 609: 815-821
- Tang C-HA, Zundell JA, Ranatunga S, Lin C, Nefedova Y, Del Valle JR, Hu C-CA (2016) Agonist-mediated activation of STING induces apoptosis in malignant B cells. *Cancer research* 76: 2137-2152
- Wu J, Chen Y-J, Dobbs N, Sakai T, Liou J, Miner JJ, Yan N (2019) STING-mediated disruption of calcium homeostasis chronically activates ER stress and primes T cell death. *Journal of Experimental Medicine* 216: 867-883
- Wu J, Liu Q, Zhang X, Tan M, Li X, Liu P, Wu L, Jiao F, Lin Z, Wu X, Wang X, Zhao Y, Ren J (2022) The interaction between STING and NCOA4 exacerbates lethal sepsis by orchestrating ferroptosis and inflammatory responses in macrophages. *Cell Death Dis* 13: 653
- Xu Y, Chen C, Liao Z, Xu P (2023) cGAS-STING signaling in cell death: Mechanisms of action and implications in pathologies. *European Journal of Immunology* 53: 2350386

Dear Jay,

Thank you for the submission of your revised manuscript to EMBO reports. We have now received the reports from both referees who support publication. Please find their reports copied below my signature. Before I can formally accept the manuscript, I need you to address some minor points below:

- Please provide up to 5 keywords.
- Please update the 'Conflict of interest' paragraph to our new 'Disclosure and competing interests statement'. For more information see <https://www.embopress.org/page/journal/14693178/authorguide#conflictsofinterest>
- The Data Availability section should only refer to data deposited in external repositories. Therefore, please remove the statement "All data of this study are included in this paper. Additional information is available upon request."
- References: et al needs to be used after 10 author names
- Please provide a complete author checklist, which you can download from our author guidelines (<<https://www.embopress.org/page/journal/14693178/authorguide>>). Please insert information in the checklist that is also reflected in the manuscript. The completed author checklist will also be part of the RPF.
- The source data for the following figure panels is missing: 2A, 2D, 2G, 3A, 3D, 4A, 4C, 4E, 4G, 4I, 6A, 6B, 6D, 6E, 6G, 7A. Please upload these and complete the Source Data Checklist.
- Source data for Fig 6G has been placed into the folder for 6C.
- Source data for the EV figures should be grouped into one zipped folder.
- The images in Figure 2D and Figure 6D appear similar. Please double check and in case some images have been re-used, please clearly state so in the figure legend.
- We can only typeset up to 5 EV figures. I checked whether you could combine some figures to reduce their number but there seems not much room for merging figures. You can put some of the figures into an Appendix. The Appendix is a single PDF file (containing figures and legends) called *Appendix*, which should start with a short Table of Content incl. page numbers. Appendix figures should be referred to in the main text as: "Appendix Figure S1, Appendix Figure S2" etc. See detailed instructions regarding expanded view here: <<https://www.embopress.org/page/journal/14693178/authorguide#expandedview>>
- Tak et al 2023 is a preprint and should be marked as that. The citation in the text is: (preprint: NAME1 et al, YEAR); in the reference list: Author NAME1, Author NAME2 (YEAR) article title. bioRxiv doi [PREPRINT].
- Our production/data editors have asked you to clarify several points in the figure legends (see below). Please incorporate these changes in the manuscript and return the revised file with tracked changes with your final manuscript submission:
 - a) Please note that a separate 'Data Information' section is required in the legends of all the figures. [The prefix 'Data Information: ' should be used for information in the legend that refers to several or all figure panels. Looking at it myself, it seems this is mainly the comment 'Bar, 10 um'. You also need to specify to which figure panels this refers to.]
 - b) Please indicate the statistical test used for data analysis in the legends of figures 3e, g, h; 4b, d, f, j; 6f, h; 7b, c, e, f, j; EV1a; EV4g; EV7b, e.
 - c) Please note that information related to n is missing in the legends of figures 7j; EV7b, e.
 - d) Please note that the error bars are not defined in the legends of figures 7j; EV7b, e
(Be careful if you calculate statistics on data obtained from cell culture. If all cells are taken from the same experiment, n is actually 1).
- Finally, EMBO Reports papers are accompanied online by A) a short (1-2 sentences) summary of the findings and their significance, B) 2-3 bullet points highlighting key results and C) a synopsis image that is 550x300-600 pixels large (width x height) in PNG for JPG format. You can either show a model or key data in the synopsis image. Please note that the size is rather small and that text needs to be readable at the final size. Please send us this information along with the revised manuscript.
- On a different note, I would like to alert you that EMBO Press offers a new format for a video-synopsis of work published with us, which essentially is a short, author-generated film explaining the core findings in hand drawings, and, as we believe, can be

very useful to increase visibility of the work. This has proven to offer a nice opportunity for exposure i.p. for the first author(s) of the study. Please see the following link for representative examples and their integration into the article web page:

<https://www.embopress.org/doi/full/10.15252/emj.2019103932>

Best wishes,

Martina

Referee #1:

Authors have addressed all my previous questions and comments. The expanded discussion and extra experimental data further strengthen and clarify the manuscript.

I believe the current manuscript to be suitable for publication in EMBO Reports.

Referee #2:

The critiques that I made for the previous Ms sent to EMBO J was mostly addressed by the authors. I am happy to accept this Ms.

All editorial and formatting issues were resolved by the authors.

Dr. Jay Xiaojun Tan
University of Pittsburgh
Cell Biology
100 Technology Drive
Suite 564
Pittsburgh, Pennsylvania 15219
United States

Dear Jay,

I am very pleased to accept your manuscript for publication in the next available issue of EMBO reports. Thank you for your contribution to our journal.

Best wishes,

Martina
